# Curcumin as an Antioxidant Against Ziprasidone Induced Lipid Peroxidation in Human Plasma: Potential Relevance to Cortico Subcortical Circuit Function

**DOI:** 10.3390/ijms262110430

**Published:** 2025-10-27

**Authors:** Anna Dietrich-Muszalska, Piotr Kamiński, Bogdan Kontek, Edward Jacek Gorzelańczyk

**Affiliations:** 1Department of Biological Psychiatry, Medical University of Łódź, 90-419 Łódź, Poland; tzn_lodz@post.pl; 2Department of Biological Psychiatry, Centre for Mental Health and Addiction, Medical University of Łódź, 90-419 Łódź, Poland; 3Division of Ecology and Environmental Protection, Department of Medical Biology and Biochemistry, Faculty of Medicine, Collegium Medicum in Bydgoszcz, Nicolaus Copernicus University in Toruń, 85-094 Bydgoszcz, Poland; 4Department of Biotechnology, Institute of Biological Sciences, Faculty of Biological Sciences, University of Zielona Góra, 65-516 Zielona Góra, Poland; 5Department of General Biochemistry, Faculty of Biology and Environmental Protection, University of Łódź, Pomorska 141/143, 90-236 Łódź, Poland; 6Institute of Philosophy, Kazimierz Wielki University in Bydgoszcz, 85-064 Bydgoszcz, Poland; medsystem@medsystem.com.pl; 7Faculty of Mathematics and Computer Science, Adam Mickiewicz University in Poznań, 61-614 Poznań, Poland; 8Medically Assisted Recovery Association “MAR”, 85-067 Bydgoszcz, Poland; 9Oskar Bielawski Greater Poland Neuropsychiatric Center in Kościan, 64-000 Kościan, Poland

**Keywords:** curcumin, antipsychotics, Ziprasidone, lipid peroxidation, oxidative stress, schizophrenia, cortico-subcortico-loops

## Abstract

Oxidative stress observed in schizophrenia and other psychiatric disorders can induce neuronal damage and modulate intracellular signaling, ultimately leading to neuronal death by apoptosis or necrosis. The aim of this study was to estimate in vitro the possible antioxidant properties of curcumin, the natural polyphenolic antioxidant, and its protective effects against lipid peroxidation induced by the atypical antipsychotic Ziprasidone. Curcumin (5 µg/mL, 12.5 µg/mL, 25 µg/mL, 50 µg/mL) was added to human plasma and incubated for 1 and 24 h, alone and in the presence of Ziprasidone (40 ng/mL, 139 ng/mL, 250 ng/mL). Control plasma samples were incubated for 1 and 24 h. The concentration of thiobarbituric acid-reactive substances (TBARSs; lipid peroxidation marker) was determined by the spectrophotometric method according to Rice-Evans. Curcumin at the tested concentrations significantly inhibited lipid peroxidation in human plasma by about 60%. Ziprasidone (40 ng/mL, 139 ng/mL, 250 ng/mL) significantly increased TBARS levels, but in the presence of the studied curcumin concentrations, its pro-oxidative effects were reduced by about 56%. Our results confirm that Ziprasidone in vitro may induce lipid peroxidation in human plasma, whereas curcumin protects against lipid peroxidation in human plasma caused by the antipsychotic Ziprasidone.

## 1. Introduction

The brain, due to its high energy demand and intense metabolic activity, is particularly susceptible to the effects of reactive oxygen species (ROS). The greatest intensity of oxidative stress is observed in structures with the highest neuronal activity, such as the striatum, which plays a key role in integrating motor, cognitive, and emotional information. Excessive production of free radicals in this region can lead to damage to lipids, proteins, and DNA, which consequently disrupts the functioning of cortico–subcortical loops, including the cortico-striato-thalamo-cortical pathways. Because of its high energy demand and metabolic activity, the brain is highly vulnerable to reactive oxygen species (ROS). Oxidative stress is particularly pronounced in the striatum, a region integrating motor, cognitive, and emotional functions, where excessive free radicals damage lipids, proteins, and DNA, thereby impairing cortico-striato-thalamo-cortical circuits. Dysfunction of these circuits is a significant element in the pathophysiology of many neurodegenerative and neuropsychiatric disorders, such as Parkinson’s disease and schizophrenia [1]. High information flow through cortico-subcortical loops, which integrate motor, cognitive, and emotional processing, may contribute to oxidative stress as a result of increased metabolic demand [2,3]. Dysfunction of cortico–subcortical circuits contributes to the pathophysiology of neurodegenerative and neuropsychiatric disorders, including Parkinson’s disease and schizophrenia [1]. Their high information flow and metabolic demand may further promote oxidative stress [2,3].

Studies have emphasized the role of cortico–subcortical loops in the regulation of cognition, emotion, and motor functions [4]. These loops, integrating the prefrontal cortex with subcortical structures such as the striatum, thalamus, and brainstem nuclei, are critical for maintaining neural predictive processing, affective regulation, and executive control [5,6]. Cortico–subcortical loops, linking the prefrontal cortex with the striatum, thalamus, and brainstem nuclei, are essential for cognition, emotion, and motor regulation, supporting predictive processing, affective control, and executive functions [4,5,6].

Redox dysregulation, reflected in decreased glutathione levels and impaired antioxidant defenses, has been observed in several cortico–subcortical regions, including the striatum, in individuals with schizophrenia. Such oxidative imbalance may increase the susceptibility of striatal neurons to lipid peroxidation, protein oxidation, and DNA damage, potentially disrupting dopaminergic signaling and cortico-striatal network function [7]. In schizophrenia, redox dysregulation with reduced glutathione and impaired antioxidant defenses has been reported in cortico–subcortical regions, including the striatum. This imbalance heightens neuronal vulnerability to oxidative damage, potentially disrupting dopaminergic signaling and cortico-striatal network function [7].

Structures of the cortico–subcortical loops, particularly the striatum, pallidum, and substantia nigra, are especially vulnerable to oxidative stress due to their high metabolic activity and elevated dopamine content, which is prone to auto-oxidation. It has been found that the loss of dopaminergic neurons in the substantia nigra pars compacta leads to a reduction in dopamine levels in the striatum, which disrupts the function of cortico–subcortical loops involving, among others, the pallidum and other basal ganglia. Cortico–subcortical structures such as the striatum, pallidum, and substantia nigra are highly vulnerable to oxidative stress due to intense metabolism and dopamine auto-oxidation. Degeneration of dopaminergic neurons in the substantia nigra pars compacta reduces striatal dopamine, thereby impairing basal ganglia loops including the pallidum.

The high metabolic activity of these structures and the presence of dopamine, which is prone to auto-oxidation, make them particularly vulnerable to oxidative damage [8]. Abundant evidence indicates that oxidative stress is associated with schizophrenia [9,10,11,12] and that some antipsychotics can increase oxidative stress and the damage of molecules [13,14,15,16]. Due to intense metabolism and dopamine auto-oxidation, these structures are highly susceptible to oxidative damage [8]. Evidence links oxidative stress to schizophrenia [9,10,11,12], and some antipsychotics may further exacerbate molecular damage by enhancing oxidative stress [13,14,15,16].

In schizophrenia, dysfunction of cortico–subcortical loops, including the prefrontal cortex, striatum, pallidum, and thalamus, may lead to increased oxidative stress through mitochondrial impairments and chronic microglial activation. These mechanisms form self-perpetuating vicious cycles in which oxidative damage exacerbates neuronal network dysfunction, which in turn further amplifies oxidative stress. As a result, the integration of signals between the cortex and subcortical structures progressively deteriorates, contributing to impairments in cognitive, emotional, and sensory integration processes [17]. In schizophrenia, dysfunction of cortico–subcortical loops involving the prefrontal cortex, striatum, pallidum, and thalamus may promote oxidative stress via mitochondrial deficits and chronic microglial activation. This creates self-perpetuating cycles of oxidative damage and neuronal dysfunction, progressively impairing cortical–subcortical integration and contributing to cognitive, emotional, and sensory deficits [17].

Dysfunction of cortico–subcortical loops (e.g., in Parkinson’s disease, schizophrenia, or depression) may exacerbate oxidative stress through mitochondrial impairments and chronic microglial activation [18]. In acute psychosis, excessive dopamine release (primarily in the mesolimbic pathway) leads to overactivation of D2 dopamine receptors [19], accompanied by increased glutamatergic activity that results in excitotoxicity [20], neuronal damage, excessive calcium influx into neurons [21], oxidative stress, and mitochondrial dysfunction [22]. Dysfunction of cortico–subcortical loops in disorders such as Parkinson’s disease, schizophrenia, and depression may intensify oxidative stress via mitochondrial deficits and microglial activation [18]. In acute psychosis, mesolimbic dopamine hyperactivity overstimulates D2 receptors [19] and enhances glutamatergic transmission, leading to excitotoxicity, calcium overload, neuronal injury, oxidative stress, and mitochondrial dysfunction [20,21,22]. There is growing scientific evidence that oxidative stress plays a pivotal role in the pathophysiology of psychoses [23,24,25]. Higher doses of antipsychotic drugs more effectively block dopamine D2 receptors and indirectly suppress excessive glutamatergic transmission, thereby reducing the risk of excitotoxicity, exerting neuroprotective effects, and stabilizing neurons in key regions (e.g., the prefrontal cortex, hippocampus, and nucleus accumbens) [23]. Accumulating evidence indicates that oxidative stress is central to the pathophysiology of psychoses [23,24,25]. At higher doses, antipsychotics more effectively block D2 receptors and suppress glutamatergic over activity, thereby reducing excitotoxicity and stabilizing neurons in regions such as the prefrontal cortex, hippocampus, and nucleus accumbens [23].

Reducing oxidative stress during acute psychosis is particularly important, as increases in oxidative stress markers (e.g., lipid peroxides, ROS) and decreases in antioxidant levels (e.g., glutathione) are observed. Certain antipsychotic drugs (e.g., clozapine, olanzapine, quetiapine) [13,26,27] possess antioxidant properties and, at higher doses, may more effectively counteract oxidative stress, thereby protecting neurons from damage (ibid). During acute psychosis, elevated oxidative markers (e.g., lipid peroxides, ROS) and reduced antioxidants such as glutathione highlight the importance of limiting oxidative stress. Some antipsychotics, including clozapine, olanzapine, and quetiapine, exhibit antioxidant properties and at higher doses may counteract oxidative damage, thereby protecting neurons [13,26,27].

Both theoretical and practical considerations highlight the importance of decisive antipsychotic treatment [28]. In neurodegenerative diseases, oxidative damage in the striatum has been closely linked to disturbances in dopamine metabolism, a key neurotransmitter within cortico–subcortical loops. The striatum, together with interconnected structures such as the pallidum and substantia nigra, is integral to these loops and is characterized by high metabolic demand and abundant dopamine content, rendering it particularly susceptible to oxidative injury through lipid peroxidation, protein oxidation, and nucleic acid damage. Such vulnerabilities may disrupt the functional integrity of cortico-striato-pallido thalamic circuits, contributing to both motor and cognitive deficits [29]. Decisive antipsychotic treatment remains essential [28]. In neurodegenerative disease, oxidative damage in the striatum disrupts dopamine metabolism within cortico–subcortical loops. Owing to high metabolic demand and dopamine content, the striatum, pallidum, and substantia nigra are especially prone to oxidative injury, which compromises cortico-striato-pallido-thalamic circuits and contributes to motor and cognitive deficits [29].

Dopaminergic signaling in the substantia nigra plays an autonomous role in regulating locomotor functions, independent of changes occurring at neuronal terminals in the striatum. The structures of the cortico-subcortical loops, including the striatum and pallidum, are particularly susceptible to disturbances resulting from dopaminergic dysfunction in the substantia nigra. The high metabolic activity of these regions, together with the presence of dopamine prone to auto oxidation, increases their vulnerability to oxidative stress, which may disrupt signal transmission throughout the loop and lead to motor deficits [30]. Dopaminergic signaling in the substantia nigra autonomously regulates locomotor functions, and its dysfunction affects cortico–subcortical loops involving the striatum and pallidum. Owing to high metabolic activity and dopamine auto-oxidation, these regions are highly vulnerable to oxidative stress, which can impair loop transmission and cause motor deficits [30].

Oxidative damage within the structures of cortico–subcortical loops may affect dopaminergic neurotransmission and the integration of signals between the cortex and the basal ganglia, which in turn influences motor control, cognitive functions, and emotions [30]. Oxidative stress develops due to the excess of ROS, a deficit in antioxidant mechanisms or a combination of both [31]. A deficit in antioxidant mechanisms can depend on a genetic defect in one or in several enzymes involved in the defense mechanisms: superoxide dismutase (SOD), catalase (CAT) or those related to the glutathione (GSH) metabolism [31]. Oxidative damage in cortico–subcortical loops can impair dopaminergic transmission and cortical–basal ganglia integration, affecting motor, cognitive, and emotional functions [30]. Oxidative stress arises from excess ROS, insufficient antioxidant defenses, or both, often linked to genetic defects in enzymes such as SOD, CAT, or those regulating glutathione metabolism [31].

Impairment of the antioxidant systems makes the organism particularly vulnerable during a temporary excess of ROS. Impairment of the antioxidant defense system leads to an excess of ROS [32]. ROS are free radical atoms or molecules with unpaired electrons such as superoxide anions, superoxide ion, highly reactive hydroxyl radical or molecules like hydroperoxide, peroxy-nitrite [33]. The main source of ROS in vivo is aerobic mitochondrial respiration, although ROS are also produced during a normal metabolic activity such as peroxisomal beta-oxidation of fatty acids. ROS are cleared from the cell by the action of the enzymes superoxide dismutase (SOD), leading to the production of hydrogen peroxide (H_2_O_2_) [34,35]. These peroxides can be reduced by catalase in peroxisomes, or by glutathione peroxidase (GPx) in the cytoplasm. These peroxides can be neutralized by enzymatic antioxidants such as catalase (CAT), superoxide dismutase (SOD), and glutathione peroxidase (GPx) in the cytoplasm [32]. ROS can also be conjugated with antioxidants, the most important being GSH [31]. Reactive oxygen species (ROS) are free radicals with unpaired electrons, including superoxide, hydroxyl radicals, hydroperoxide, and peroxy-nitrite [33]. Their primary source is mitochondrial respiration, though they also arise from processes such as peroxisomal β-oxidation. ROS are detoxified by SOD, which generates H_2_O_2_ subsequently reduced by catalase or glutathione peroxidase, and by conjugation with antioxidants, chiefly glutathione [31,34,35].

When natural defense mechanisms of the organism are efficient, ROS exist in balance with biochemical antioxidants and detoxifying enzymes [33]. When this balance is changed in favor of ROS, oxidative stress occurs resulting in a generation of oxidatively modified molecules and disruption of normal cellular activity [31]. The main damage to the cells results from the interaction of ROS with molecules within the cell, proteins, lipids and DNA [36,37]. The main damage to the cells caused by ROS involves proteins, lipids, and DNA [32]. The effects of oxidative stress depend upon the size of the induced changes. More severe oxidative stress can cause cell death and even moderate oxidation can trigger the apoptosis or necrosis [38,39]. Under normal conditions, ROS are balanced by antioxidants and detoxifying enzymes [33]. Excess ROS disrupt this balance, modifying proteins, lipids, and DNA, and impairing cellular function [31,36,37]. While moderate oxidative stress may trigger apoptosis or necrosis, severe stress can lead to cell death [38,39]. Thanks to its thiol group, GSH can scavenge free radicals and detoxify the potential harmful xenobiotics as well as their metabolites [40]. Increasing evidence suggests that oxidative stress mediates neuropathological processes in neuropsychiatric disorders and is involved in the etiology and progression of major psychiatric disorders such as schizophrenia, major depression, bipolar disorder [41,42,43,44]. Through its thiol group, glutathione (GSH) neutralizes free radicals and detoxifies xenobiotics [40]. Growing evidence indicates that oxidative stress contributes to neuropathology and progression of major psychiatric disorders, including schizophrenia, depression, and bipolar disorder [41,42,43,44].

The contribution of oxidative injury to the pathophysiology of schizophrenia has been indicated by the increase in lipid peroxidation products in plasma, platelets and erythrocytes, and cerebrospinal fluid [45], and the altered levels of both enzymatic and non-enzymatic antioxidants in chronic, naïve, and first-episode patients [10,45,46,47,48,49,50]. The measurement of isoprostanes is useful for monitoring the effectiveness of schizophrenia treatment [51]. In schizophrenia, oxidative injury is reflected by elevated lipid peroxidation in plasma, platelets, erythrocytes, and cerebrospinal fluid [45] and by altered enzymatic and non-enzymatic antioxidant levels across chronic, naive, and first-episode patients [10,45,46,47,48,49,50]. Isoprostane measurement provides a useful marker for monitoring treatment efficacy [45].

The first study on the assessment of the increased F2-isoprostane concentration (8-iso-PGF 2 α) in patients with schizophrenia was the first one showing that the oxidation of arachidonic acid occurs through the nonenzymatic pathway (not associated with cyclooxygenase), as a result of free radical attack on membrane structures [45]. Antipsychotics are one of the most commonly used drugs to calm, sedate and control symptoms of mania as well as relieve positive symptoms of schizophrenia [52]. Oxidative stress in psychiatric disorders, manifested by specific markers such as increased lipid peroxidation, can also be caused by treatment of patients with some antipsychotic drugs and the prooxidant action of some antipsychotics which has been described [16,53]. To estimate lipid peroxidation level the assay with thiobarbituric acid is commonly used [54]. The first evidence of elevated F2-isoprostanes (8-iso-PGF_2_α) in schizophrenia showed that arachidonic acid can undergo non-enzymatic oxidation via free radical attack, independent of cyclooxygenase [51]. Antipsychotics are used to control mania and positive symptoms of schizophrenia [52], yet some possess pro-oxidant properties that enhance lipid peroxidation and oxidative stress [16,53]. Lipid peroxidation is typically assessed with the thiobarbituric acid assay [54].

Our earlier studies presented that haloperidol in vitro causes an increase in thiobarbituric acid-reactive substances (TBARSs) in human plasma, contrary to the action of risperidone and olanzapine that have no such effect on the level of TBARS in plasma [55,56]. The same effects were observed when human plasma was administered with other, second-generation antipsychotics: amisulpride, clozapine, or quetiapine [57,58]. Our earlier study demonstrated that Ziprasidone induces increased lipid peroxidation in plasma estimated by the level of TBARS in plasma [59]. It has been demonstrated that haloperidol increases thiobarbituric acid-reactive substances (TBARSs) in human plasma, unlike risperidone and olanzapine, which do not affect TBARS levels [55,56]. Similar results were reported for other second-generation antipsychotics, including amisulpride, clozapine, and quetiapine [57,58]. In contrast, Ziprasidone was shown to increase lipid peroxidation, as indicated by elevated TBARS in plasma [59].

Therefore, the aim of our study was to establish the antioxidant effects of curcumin as a potential therapeutic agent that may prevent lipid peroxidation induced by Ziprasidone action in human plasma in vitro. Oxidative stress and oxidative/nitrative damage to various molecules in schizophrenia and other psychiatric disorders may be dependent partly on the used antipsychotics [60]. This study aimed to assess the antioxidant effects of curcumin as a potential therapeutic agent against Ziprasidone-induced lipid peroxidation in human plasma in vitro. Oxidative and nitrative damage in schizophrenia and related psychiatric disorders may partly depend on the antipsychotics used [60]. Lipid peroxidation is considered to be an important marker of oxidative stress and it is found to be increased after treatment with antipsychotics, especially with first-generation antipsychotics [16,23,25,61]. Lipid peroxidation, a key marker of oxidative stress, is elevated following antipsychotic treatment, particularly with first-generation agents [16,23,25,61]. In the prophylaxis and treatment of various diseases including schizophrenia, plant-derived dietary compounds (polyphenols) with anti-inflammatory and antioxidative properties may play an important role [62,63,64]. Our earlier study showed that polyphenols derived from berries of Black chokeberry *Aronia melanocarpa* reduce plasma lipid peroxidation induced by antipsychotic Ziprasidone which has pro-oxidative properties [59]. Plant-derived polyphenols with anti-inflammatory and antioxidant properties may support the prevention and treatment of disorders such as schizophrenia [62,63,64]. Our previous work demonstrated that polyphenols from Black chokeberry *Aronia melanocarpa* reduce Ziprasidone-induced plasma lipid peroxidation [59].

Ziprasidone is a benziso-thiazolyl piperazine derivative and it is the atypical antipsychotic drug from a class of agents that is used to treat a variety of psychoses because of their superiority with regard to extrapyramidal symptoms [65]. Ziprasidone targets a broad spectrum of schizophrenia symptoms, including positive, negative, and affective symptoms, with minimal motor, cognitive, prolactin-related, or anticholinergic side effects [66]. Ziprasidone has high affinity for dopamine D2 receptors and 5-HT receptors. It is a 5-HT1A receptor agonist and an antagonist for 5-HT2A, 5-HT2C and 5-HT1B/1D receptors and a moderately potent inhibitor of norepinephrine and 5-HT uptake transporters [67,68]. Ziprasidone, a benziso-thiazolyl-piperazine derivative, is an atypical antipsychotic effective across positive, negative, and affective symptoms of schizophrenia, with fewer extrapyramidal, cognitive, prolactin-related, or anticholinergic side effects [65,66]. It shows high affinity for dopamine D2 and serotonin receptors, acting as a 5-HT1A agonist, an antagonist at 5-HT2A, 5-HT2C, and 5-HT1B/1D receptors, and a moderate inhibitor of norepinephrine and serotonin reuptake [67,68].

The unique pharmacological profile of Ziprasidone may be related to a low propensity for extrapyramidal side effects, cognitive deficits and weight gain [66,67,68,69,70]. Due to the fact that Ziprasidone has a tendency toward pro-oxidative activity [71], the original study attempted to determine the potential protective role of curcumin against lipid peroxidation induced by Ziprasidone. The multiple beneficial effects of curcumin can be linked to its anti-inflammatory and antioxidative activities in vivo and in vitro [72]. Curcumin, diferuloylmethane [1,7-bis(4-hydroxy-3-methoxyphenyl)-1,6-heptadiene-3,5-dione], is a polyphenol extracted from the turmeric, powdered rhizome of Turmeric *Curcuma longa* [73]. This yellow-colored spice is widely used by Asians in the preparation of curry [74]. Ziprasidone’s pharmacological profile is associated with a low risk of extrapyramidal symptoms, cognitive deficits, and weight gain [66,67,68,69,70]. However, its pro-oxidative activity [71] prompted investigation of curcumin as a potential protective agent against Ziprasidone-induced lipid peroxidation. Curcumin (diferuloylmethane), a polyphenol from the rhizome of Turmeric *Curcuma longa*, exhibits anti-inflammatory and antioxidant effects in vivo and in vitro [72,73] and is widely used as a yellow spice in curry preparation [74].

Over the past three decades, curcumin has been the subject of thousands of papers studying its antioxidant, anti-inflammatory, cancer chemo-preventive and potentially chemotherapeutic properties [75,76,77,78]. Moreover, curcumin seems to possess antidepressant activities and could enhance the treatment of depressive symptoms [79,80]. Curcumin, a polyphenolic compound derived from Turmeric *Curcuma longa*, exhibits neuroprotective effects partly by counteracting neuroleptic-induced oxidative damage and preserving mitochondrial function [81]. Curcumin modulates critical cortico–subcortical pathways involved in motor and cognitive functions. In a rat model of Parkinson’s disease (6-OHDA), oral administration of curcumin protected dopaminergic neurons in the substantia nigra and their terminals in the striatum through activation of the α7 nicotinic acetylcholine receptor (α7 nAChR), improving motor behavior [82,83]. Over the past three decades, curcumin has been extensively studied for its antioxidant, anti-inflammatory, chemo-preventive, and potential chemotherapeutic properties [75,76,77,78]. It also shows antidepressant activity and may enhance treatment of depressive symptoms [79,80]. As a polyphenol from Turmeric *Curcuma longa*, curcumin exerts neuroprotective effects by counteracting neuroleptic-induced oxidative damage and preserving mitochondrial function [81]. It modulates cortico-subcortical pathways relevant to motor and cognitive functions, and in a rat Parkinson’s model (6-OHDA), oral curcumin protected dopaminergic neurons via α7 nicotinic acetylcholine receptor activation, improving motor behavior [82,83]. In aging mice, curcumin enhanced dendritic spine density in the prefrontal cortex and hippocampal CA3 region, improved spatial memory, and reduced oxidative stress markers [84]. In aging mice, curcumin increased dendritic spine density in the prefrontal cortex and hippocampal CA3, improved spatial memory, and lowered oxidative stress markers [84].

Electrophysiological studies revealed curcumin attenuates long-term depression (LTD), upregulates brain-derived neurotrophic factor (BDNF), and decreases activity of cyclooxygenase-2 (COX-2) expression, restoring synaptic function following chronic stress [85]. Furthermore, curcumin normalizes hypothalamic–pituitary–adrenal (HPA) axis dysregulation and restores serotonergic neurotransmission by modulating 5-HT1A and 5-HT4 receptors, activating the cAMP-PKA-CREB pathway to promote synaptic plasticity [86,87]. Electrophysiological studies show that curcumin attenuates long-term depression, upregulates BDNF, and reduces COX-2 expression, thereby restoring synaptic function after chronic stress [85]. It also normalizes HPA axis activity and enhances serotonergic neurotransmission through 5-HT1A and 5-HT4 receptor modulation, activating the cAMP-PKA-CREB pathway to support synaptic plasticity [86,87]. Nanocurcumin, a nanoparticle formulation with enhanced bioavailability and blood–brain barrier penetration, shows superior efficacy in modulating cortico–subcortical circuits. In the chronic stress models, nanocurcumin elevated BDNF and serotonin levels, improved hippocampal synaptic plasticity, and exhibited antidepressant-like effects [88]. It also prevents memory impairment and hippocampal apoptosis via activation of intracellular Akt and CaMKII-α pathways, crucial for neuronal survival [89]. Nanocurcumin, a nanoparticle formulation with improved bioavailability and brain penetration, shows enhanced efficacy in modulating cortico–subcortical circuits. In chronic stress models, it increased BDNF and serotonin levels, improved hippocampal plasticity, and produced antidepressant-like effects [88]. It also prevented memory impairment and hippocampal apoptosis through activation of Akt and CaMKII-α pathways essential for neuronal survival [89]. Additionally, nanocurcumin reduces oxidative stress and inflammation in lipopolysaccharide (LPS)-induced neuroinflammatory models, preserving hippocampal and subcortical integrity [90]. Taken together, these findings support the therapeutic potential of curcumin and nanocurcumin in restoring cortico–subcortical loop function, alleviating neuropsychiatric symptoms, and protecting against neuroleptic-induced neurotoxicity [91]. The present study aimed to characterize antioxidant action of curcumin at different concentrations in vitro on plasma lipid peroxidation caused by Ziprasidone, a second-generation antipsychotic with pro-oxidant properties. Nanocurcumin reduces oxidative stress and inflammation in LPS-induced neuroinflammatory models, preserving hippocampal and subcortical integrity [90]. Collectively, these findings highlight the therapeutic potential of curcumin and nanocurcumin in restoring cortico–subcortical function, alleviating neuropsychiatric symptoms, and protecting against neuroleptic-induced neurotoxicity [91]. The present study therefore aimed to evaluate the antioxidant effects of curcumin at different concentrations in vitro on plasma lipid peroxidation induced by Ziprasidone, a second-generation antipsychotic with pro-oxidant properties.

This study has several limitations. Firstly, the experiments were conducted in vitro using human plasma, which does not fully replicate the complexity of in vivo conditions. Secondly, we did not assess long-term effects or potential interactions with other biological pathways that may influence oxidative stress in clinical settings. Thirdly, the study focused on lipid peroxidation as the primary marker of oxidative damage, without including additional biomarkers that could provide a broader picture of redox imbalance. Finally, while curcumin was tested at different concentrations, pharmacokinetic aspects such as bioavailability and metabolism were not addressed. These limitations should be considered when interpreting the findings and highlight the need for further in vivo and clinical studies.

The novelty of this study lies not only in demonstrating the protective effects of curcumin against Ziprasidone-induced lipid peroxidation in human plasma, but also in interpreting these findings from the perspective of cortico-subcortical loop functioning. This approach provides a broader neurobiological context, linking oxidative stress with dopaminergic neurotransmission and the integration of cortical basal ganglia circuits, thereby offering new insights into the mechanisms underlying neuropsychiatric disorders.

## 2. Results

The present in vitro study demonstrates that curcumin at all tested concentrations (5 µg/mL, 12.5 µg/mL, 25 µg/mL, 50 µg/mL) significantly reduced TBARS level in human plasma (after 1 h incubation *p* < 0.03; *p* < 10^−3^; *p* < 10^−3^; *p* < 0^−3^, respectively, and after 24 h incubation for all tested concentrations *p* < 10^−6^) (Figure 1A–C). The inhibition of TBARS production was dose dependent. The antioxidant effects of curcumin were observed after 1 h as well as 24 h incubation. The strong inhibitory effect of curcumin was shown already for its lowest studied concentration. Curcumin at the concentrations used in the study reduced lipid peroxidation in plasma by 60% on average (Figure 1A–C).

The exposure of samples of human plasma to 40, 139, 250 ng/mL Ziprasidone alone resulted in an increase in lipid peroxidation measured as TBARS level in samples after 1 and 24 h incubation (in comparison to control samples without drug *p* < 0.05). The pro-oxidative effect of Ziprasidone was reduced in the presence of curcumin (Figure 2, Figure 3 and Figure 4; Table 1 and Table 2).

The presence of curcumin at all used concentrations suppressed the formation of TBARS in plasma samples incubated with Ziprasidone in a dose-independent manner.

## 3. Discussion

This study focused on a specific antipsychotic agent and utilized plasma from healthy donors, which provided valuable mechanistic insights but limits the broader applicability of the findings. Extrapolation to other antipsychotics and to plasma derived from psychiatric patients would enhance the generalizability and translational relevance of these observations. Antipsychotic drugs differ substantially in their pharmacodynamic profiles, oxidative burden, and metabolic side effects [92,93], and psychiatric populations often present with altered redox homeostasis and inflammatory markers [94,95]. Future studies should therefore incorporate diverse antipsychotic classes and patient-derived biological samples to better capture clinical heterogeneity.

Ziprasidone is an atypical antipsychotic drug whose mechanism of action involves simultaneous antagonism of dopamine D_2_ receptors and serotonin 5-HT_2_A receptors, as well as partial agonism of 5-HT_1_A receptors [96]. By blocking D_2_ receptors in the mesolimbic pathway, it reduces excessive dopaminergic activity within the limbic loop, which translates into a reduction in positive symptoms of schizophrenia. Concurrently, antagonism of 5-HT_2_A receptors in the prefrontal cortex enhances dopamine release in the striatum and cortex, potentially improving cognitive functions and lowering the risk of extrapyramidal symptoms [97]. Ziprasidone’s influence on cortico–subcortical associative and limbic loops stabilizes signal integration between cortical and subcortical structures, thereby supporting better regulation of emotion, behavior, and executive processes in patients with psychotic disorders [96,98]. Ziprasidone, as an atypical antipsychotic drug, exerts a complex influence on the oxidative–antioxidative balance, which may be relevant to the pathophysiology of schizophrenia and other psychiatric disorders. In vitro studies indicate that the drug may enhance lipid peroxidation in plasma. Leading to elevated levels of ROS and damage to cellular membranes [59]. This mechanism is likely associated with impaired mitochondrial respiratory chain function and modulation of antioxidant enzyme activity, including superoxide dismutase and glutathione peroxidase [9].

The discussion appropriately links the findings to dopaminergic and antioxidant pathways, which are well established in the pathophysiology of neuropsychiatric disorders [17,23]. Curcumin has been repeatedly shown to exert antioxidant and neuroprotective effects, including modulation of BDNF, COX-2, and serotonergic signaling [85,87]. These properties support its potential as an adjunct therapy in schizophrenia and related conditions [91,99].

The main limitation of the present study is the reliance on plasma samples, which lack cellular and immune components. While plasma-based TBARS assays are a validated tool for assessing lipid peroxidation [54,55], they cannot capture the complexity of oxidative stress in neuronal or glial cells, Therefore, further studies in cellular and animal models are warranted [42].

F2-isoprostanes are considered a highly specific and reliable biomarker of lipid peroxidation [100,101]; however, it should be emphasized that their measurement is most commonly performed in urine samples, where they serve as an indicator of overall oxidative stress intensity. In contrast, the present original study was designed to assess oxidative stress directly in plasma, where the TBARS method remains a widely applied and practical technique [102]. Plasma was selected as the biological matrix due to its clinical availability, the feasibility of repeated sampling, and its relevance for monitoring circulating oxidative stress markers in the studied population. Although the TBARS method has limitations in terms of specificity, it represents a feasible and well-established approach for plasma analysis, particularly in clinical and translational settings where urine samples are not always available or appropriate. The use of TBARS in this specific study therefore reflects the adequacy of this method for plasma measurements [102], while simultaneously acknowledging that F2-isoprostanes would be the preferred biomarker in studies based on urine samples [103].

Another important issue concerns curcumin’s bioavailability and stability. Curcumin is known to degrade rapidly under physiological conditions and shows poor systemic absorption [104,105]. Novel formulations such as nanocurcumin have been proposed to overcome these limitations [106]. Moreover, potential interactions with antipsychotics should be considered, as antipsychotic drugs themselves can modulate oxidative stress pathways [15,49]. Overall, the study provides preliminary but valuable evidence that curcumin can attenuate Ziprasidone-induced oxidative damage. However, future work should address dosing strategies, mechanistic pathways, and translational challenges before clinical application can be recommended [78].

In the context of cortico–subcortical loops, excessive oxidative stress may disrupt signal transmission between cortical and subcortical structures, affecting cognitive and emotional functions. Notably, polyphenols, such as those derived from Black chokeberry *Aronia melanocarpa* have been shown to mitigate the pro-oxidative effects of Ziprasidone, suggesting potential benefits of adjunctive antioxidant therapies [59]. Expanding the use of optimal antioxidant strategies, including supplementation with ascorbic acid [107] and E, N-acetylcysteine, or plant-derived polyphenols, may help counteract these adverse processes, support neuroprotection, and enhance cognitive function and overall treatment efficacy [9,59,96].

In this study, we demonstrate that curcumin effectively attenuates lipid peroxidation induced by the second-generation antipsychotic neuroleptic Ziprasidone. These findings add to the growing body of evidence supporting the antioxidant potential of curcumin and suggest its possible role in mitigating oxidative stress associated with antipsychotic therapy. It has been demonstrated that curcumin exerts protective effects against oxidative and nitrative changes in plasma proteins and lipids. It has been found to inhibit the formation of carbonyl groups in plasma and blood platelet proteins induced by peroxy-nitrite, a strong oxidant [108]. It inhibits carbonyl group formation in plasma and blood platelet proteins induced by peroxynitrite, a strong oxidant. The dose of 50 µg/mL curcumin inhibited the formation of carbonyl group even by 40% [108]. The antioxidant activity of curcumin against lipid peroxidation caused by peroxynitrite was also observed.

Curcumin suppressed the generation of TBARS in blood platelets and in human plasma. The concentration of 50 µg/mL of curcumin decreased the TBARS level in human plasma by about 35% [108]. The presented findings from original research indicate that curcumin reduces plasma lipid peroxidation by approximately 60% at the tested concentrations. The multiple beneficial effects of curcumin could be linked to its antioxidative and anti-inflammatory properties in in vivo and in vitro models. Curcumin, an orange-yellow component of turmeric or curry powder, is a polyphenol natural plant product isolated from the Turmeric *Curcuma longa*, commonly known as Turmeric, a tropical plant native to Asia. Curcumin, a hydrophobic polyphenol is a principal active component of Turmeric [78]. Curcumin is a highly pleiotropic molecule with a safety profile targeting multiple diseases (among others cancer, diabetes, hypertension. Alzheimer’s disease, and Parkinson disease) on the molecular level and strong evidence supports its properties [82,105,109]. It down-regulates proinflammatory cytokine expression such as tumor necrosis factor (TNF-α), interleukins IL 1, IL 2, IL 6, IL 8, IL 12 and chemokines, probably due to inactivation of the nuclear transcription factor NF κB [110,111].

Although our study demonstrated a reduction in lipid peroxidation (TBARS), it did not investigate the molecular pathways through which curcumin counteracts Ziprasidone-induced oxidative stress, such as Nrf2 activation, modulation of antioxidant enzymes, or glutathione regulation. While TBARS measurement provided a useful indication of lipid peroxidation, we recognize that it does not fully elucidate the molecular pathways through which curcumin mitigates Ziprasidone-induced oxidative stress. Curcumin has been shown to exert antioxidant effects via multiple mechanisms, including activation of the Nrf2 signaling pathway, enhancement of antioxidant enzyme activities such as superoxide dismutase (SOD) and catalase (CAT), and modulation of glutathione (GSH) levels [93,94,95]. These pathways are particularly relevant in the context of antipsychotic-induced oxidative burden, which may vary depending on drug class and patient-specific factors. We have added a statement to the revised manuscript acknowledging this limitation and recommending that future studies incorporate assessments of Nrf2 protein expression, enzymatic antioxidant activity, and GSH content to provide deeper mechanistic insight.

Anti-inflammatory effects of curcumin are connected with the inhibition of the arachidonic acid pathway and suppression of cyclooxygenase 2 (COX 2) and prostaglandin PGE_2_ synthesis. It is able to inhibit the induction of COX 2 gene expression in epithelial cells [112]. Curcumin is known to reduce colonic nitrate level, it down regulates nitric oxide synthase (iNOS) and inhibits activation of p38 mitogen-activated protein kinase (MAPK) [113]. The underlying mechanisms of numerous effects of curcumin are diverse and appear to involve the regulation of different molecular targets. Curcumin targets transcription factors, growth factors, protein kinases, inflammatory cytokines, adhesion molecules, apoptosis-related proteins, enzymes (PLA_2_, COX, LOX) and many others [114]. As a highly pleiotropic molecule, curcumin may interact with different targets. It has been shown that more than 30 different proteins can directly interact with curcumin, including thioredoxin reductase [115], focal adhesion kinase [116], protein kinase C PKC [117,118], DNA polymerase [119], lipoxygenase [120], tubulin [121] and divalent metal ions [122]. Curcumin is unstable at basic pH, but in the presence of human blood less than 20% of curcumin was found to be decomposed in 1 h [104].

Our results in vitro confirmed the antioxidant activity of curcumin in plasma after 1 and 24 h treatment (Figure 1, Figure 2, Figure 3 and Figure 4, Table 1 and Table 2). Preclinical studies suggest that curcumin can inhibit the induction of macrophage NOS activation at the concentration of 1–20 µM [123]. Curcumin showed a dose- and time-dependent induction of heme oxygenase 1 gene at the concentration of 1–8 µM via the NF κB pathway [124]. Curcumin is a potent antioxidant due to its ability to directly scavenge ROS: O_2_•^−^, •OH, NO and peroxy-nitrite that is formed in rapid reaction of NO with superoxide anion [125]. Preclinical antioxidative studies reveal that curcumin at the concentration of 10 µM inhibited ROS generation in rat macrophages [126] and in red blood cells [127]. It was shown that curcumin scavenges O_2_•^−^ and •OH radicals [128]. In a randomized, placebo-controlled study in patients with chronic tropical pancreatitis who were administered 500 mg of oral curcumin and 5 mg of piperazine for 6 weeks, it was shown that patients who received curcumin (with piperazine) had significantly lower levels of MDA, a marker of lipid peroxidation [129].

The concentrations of curcumin applied in the present in vitro experiments (typically in the range of several to tens of µg/mL) are substantially higher than those achievable in human plasma after oral administration of conventional curcumin supplements. Pharmacokinetic studies consistently demonstrate that, due to poor solubility, rapid metabolism, and extensive conjugation, plasma levels of free curcumin rarely exceed the low nanomolar range (1–20 ng/mL, i.e., <0.05 µM) even after gram-level oral dosing [105,130,131]. For example, in a real-world cohort, unconjugated plasma curcumin concentrations after supplementation remained between 1.0 and 18.6 ng/mL, which is several orders of magnitude below the micromolar concentrations used in most in vitro assays [131]. Formulation strategies such as nanoparticles, liposomal carriers, or water-soluble complexes can markedly enhance systemic exposure. A recent crossover trial with a water-soluble formulation (AQUATURM^®^) reported an upper 7-fold increase in plasma exposure compared to standard curcumin tablets, achieving peak concentrations in the range of 100–300 ng/mL [132]. While the µg/mL levels used in vitro are still low, such formulations demonstrate that higher systemic concentrations are technically achievable.

With respect to safety, curcumin has been generally recognized as safe in humans at oral doses up to 8 g/day, with only mild gastrointestinal side effects reported [133]. Thus, while the in vitro concentrations tested in this study are not directly attainable in plasma under standard supplementation, they provide a mechanistic proof-of-concept. Translational relevance requires cautious interpretation, and future work should explore whether optimized formulations or local tissue accumulation (e.g., in the gastrointestinal tract or brain with nanocurcumin) can bridge the gap between in vitro efficacy and clinically achievable exposure.

No side effects have been reported by patients during or after the curcumin treatment. Curcumin was shown to be neuroprotective against ethanol-induced brain injury in vivo following oral administration [128]. This effect was related to a reduction in lipid peroxidation and enhancement of GSH in the brain [134]. Curcumin, due to its ability to neutralize ROS and RNS, inhibit lipid peroxidation, and modulate the expression of proinflammatory cytokines, may support the proper functioning of cortico–subcortical loops, Its action on signaling pathways such as NF κB, MAPK, and Nrf2 may limit neuro-inflammatory neuronal damage and improve synaptic transmission within the cortex and basal ganglia [135,136,137]. As a result, this may promote better integration of neuronal signals and a reduction in symptoms resulting from dysfunction of these circuits. Preclinical studies indicate that curcumin may be a valuable adjunct in anti-psychotic treatment for schizophrenia. Owing to its antioxidant, anti-inflammatory, and neuroprotective properties, curcumin may alleviate negative and cognitive symptoms, as well as reduce adverse effects of neuroleptics such as movement disorders and dyslipidemia. The mechanisms underlying these effects include, among others, the reduction of oxidative stress, modulation of proinflammatory cytokine expression, and improvement of neuronal cell membrane integrity. Importantly, in clinical studies to date, curcumin has been well tolerated and has not caused significant side effects, making it a promising candidate for further investigation as an adjunctive therapy in schizophrenia [138].

Cortico–subcortical loops, comprising connections between the prefrontal cortex, striatum, globus pallidus, thalamus, and limbic structures, play a key role in integrating motor, cognitive, and emotional information. Their proper functioning is essential for maintaining the balance between neuronal excitation and inhibition, as well as for coordinating executive processes and mood regulation. In psychotic disorders such as schizophrenia, disturbances in the activity of these circuits are observed, which may result from excessive oxidative stress and chronic inflammation. Reactive oxygen species (ROS) and reactive nitrogen species (RNS) damage neurons within the cortex and basal ganglia, leading to impairments in dopaminergic, glutamatergic, and GABAergic neurotransmission. Such changes can result in abnormal synchronization of neuronal activity, reduced synaptic plasticity, and impaired signal integration within cortico–subcortical loops, which in turn contribute to the exacerbation of cognitive, negative, and motor symptoms. From this perspective, interventions with antioxidant and anti-inflammatory properties, such as curcumin supplementation, may represent an important strategy for protecting the function of these circuits and supporting the effectiveness of antipsychotic treatment [17].

Curcumin may act as a positive allosteric modulator of the α7 nicotinic acetylcholine receptor (α7 nAChR), enhancing currents induced by receptor agonists and improving cholinergic transmission. Animal models have demonstrated that such effects can alleviate social deficits and reduce oxidative stress in the brain, which is relevant in the context of cognitive and psychiatric disorders [139]. The α7 nicotinic acetylcholine receptor (α7 nAChR) plays an important role within the striatum, the main subcortical structure of the cortico–subcortical loops, influencing neurotransmission mechanisms and neuroprotective processes. Activation of the α7 nAChR has been found to modulate dopamine release, which is relevant for regulating motor functions and reward processing in cortico–subcortical loops. The α7 nAChR also contributes to reducing oxidative stress and enhances synaptic plasticity by increasing calcium influx into neurons. Due to these properties, this receptor is considered a promising therapeutic target in the treatment of neurodegenerative disorders, including Parkinson’s disease [140]. It has been found that in rat cortical neurons, curcumin protects cells against glutamate induced excitotoxicity, partly by increasing BDNF levels and activating the TrkB receptor. This effect may stabilize glutamatergic transmission and limit synaptic damage under conditions of oxidative stress [141].

One of the major limitations of curcumin is its low oral bioavailability, which results from poor aqueous solubility, instability at physiological pH, rapid intestinal and hepatic metabolism, and extensive systemic elimination [105,142]. After conventional oral administration, plasma concentrations of free curcumin typically remain in the low nanomolar range, far below the micromolar levels often used in in vitro assays [130,131]. This pharmacokinetic profile underscores the translational challenge of extrapolating in vitro findings directly to clinical settings. To overcome these limitations, several delivery strategies have been developed. Co-administration with piperine has been shown to inhibit glucuronidation and enhance systemic exposure [105]. More advanced approaches include nanoparticle formulations. Liposomes, micelles, and phospholipid complexes improve solubility, stability, and tissue penetration [81]. Clinical studies with water-soluble formulations such as AQUATURM^®^ have demonstrated significantly higher plasma concentrations compared to conventional tablets, although they are still below the levels used in vitro [132]. By acknowledging these pharmacokinetic barriers and highlighting ongoing formulation advances, the translational context of our findings is strengthened. Future studies should evaluate whether optimized delivery systems can achieve therapeutically relevant concentrations in target tissues, thereby bridging the gap between mechanistic in vitro evidence and clinical applicability.

Brain-derived neurotrophic factor (BDNF) is one of the main regulators of neuronal survival, dendritic growth, and the formation of new synapses. Within the cortico–subcortical loops, which include the prefrontal cortex, striatum, globus pallidus, and thalamus, an adequate level of BDNF is essential for maintaining synaptic plasticity and the adaptive modulation of signals [143]. The striatum receives numerous glutamatergic projections from the cerebral cortex, and excessive activation of N-methyl D-aspartate (NMDA) receptors and α-amino-3-hydroxy-5-methyl 4-isoxazole-propionic acid (AMPA) receptors by glutamate leads to Ca^2+^ influx, oxidative stress, and neuronal death, particularly of medium spiny neurons in the striatum. By reducing glutamate induced excitotoxicity through increasing BDNF levels and activating TrkB, curcumin can stabilize glutamatergic transmission, which may protect striatal neurons from degeneration [141].

The chemistry of curcuminoids is closely related to the broader class of diarylheptanoids, which share a characteristic 1.7-diphenylheptanoid backbone. A recent comprehensive review has highlighted advances in the synthetic strategies, structural diversity, and pharmacological potential of diarylheptanoids, providing valuable context for understanding curcumin and its analogues [144]. Incorporating this perspective strengthens the chemical framework of our study, as it underscores how structural modifications of the diarylheptanoid scaffold can influence antioxidant, anti-inflammatory, and neuroprotective activities. This broader chemical insight complements our findings and situates curcumin within a larger family of bioactive natural products with translational relevance in oxidative stress research [105,142].

Preclinical studies indicate that curcumin may modulate the serotonergic and dopaminergic systems by influencing the density and sensitivity of 5 HT_1_A, 5 HT_2_A, and dopamine D_2_ receptors. This mechanism may be related to its antidepressant effects and its potential to support antipsychotic therapy [145]. Curcumin may exert neuroprotective effects through modulation of the GABAergic system. In a rat model of spinal cord injury, administration of curcumin reduced oxidative stress and increased the expression of GABA_A_A_ receptors and the enzyme glutamate decarboxylase 65 (GAD65), which was associated with decreased pain perception [146]. This mechanism may also be relevant in the context of neurodegenerative diseases in which the balance between neuronal excitation and inhibition is disrupted. Curcumin can modulate GABA_A_A_ receptor activity, enhancing postsynaptic inhibition and exhibiting anxiolytic effects in animal models. This effect may be important for maintaining excitation–inhibition balance in the brain [146].

Curcumin may exert neuroprotective effects through modulation of the GABAergic system. In a rat model of spinal cord injury, administration of curcumin reduced oxidative stress and increased the expression of GABAA__A_ receptors and the enzyme glutamate decarboxylase 65 (GAD65), which was associated with decreased pain perception [146]. This mechanism may also be relevant in the context of neurodegenerative diseases in which the balance between neuronal excitation and inhibition is disrupted. Curcumin can modulate GABAA__A_ receptor activity, enhancing postsynaptic inhibition and exhibiting anxiolytic effects in animal models. This effect may be important for maintaining excitation–inhibition balance in the brain [146].

A key aspect is the discussion of potential pharmacological interactions between curcumin and antipsychotic drugs, as this helps to better contextualize the translational relevance of our findings, Curcumin has been reported to modulate cytochrome P450 enzymes and drug transporters, which may theoretically alter the pharmacokinetics of antipsychotics such as Ziprasidone, risperidone, or clozapine [92,93]. In addition, curcumin’s antioxidant and anti-inflammatory properties may attenuate some adverse effects of antipsychotics, including oxidative stress, metabolic disturbances, and extrapyramidal symptoms [93]. However, clinical data on direct interactions remain limited, and most evidence derives from preclinical or mechanistic studies. Importantly, drug-interaction databases list multiple moderate interactions between curcumin and commonly prescribed medications, underscoring the need for careful monitoring in polypharmacy contexts [147]. Therefore, while curcumin shows promise as an adjunctive therapy in schizophrenia, further pharmacokinetic and clinical studies are required to establish the safety of combined use with antipsychotics, particularly Ziprasidone.

Evaluating additional endpoints, including intracellular reactive oxygen species (ROS) levels and protein oxidation, would provide deeper mechanistic insight into the antioxidant effects observed. Although these parameters were beyond the scope of the present study, we recognize their relevance and have included a statement in the revised manuscript encouraging their assessment in future investigations. While our current approach focused on plasma-based oxidative markers, future investigations incorporating cellular assays would allow for a more comprehensive understanding of redox modulation at the molecular level. Such endpoints are widely recognized for their relevance in assessing oxidative stress and therapeutic efficacy in vitro and in vivo [94,95,148]. We have included a statement in the revised manuscript to reflect this perspective and encourage further exploration using cell-based or tissue-specific models.

Preclinical data suggest that curcumin may modulate the serotonergic and dopaminergic systems by influencing the density and sensitivity of 5 HT_1A_, 5 HT_2A_, and D_2_ receptors. This mechanism may be related to its antidepressant effects and its potential to support antipsychotic therapy [145]. Curcumin may modulate GABA_A_A_ receptor activity, enhancing postsynaptic inhibition and exhibiting anxiolytic effects in animal models. This effect may be relevant in the context of regulating the excitation–inhibition balance in the brain [149]. Curcumin was shown to increase GSH in different cell lines [150,151]. This increase in GSH level caused by curcumin could be due to an increase in de novo GSH synthesis following the increased activity of glutamate cysteine ligase owing to an induction of its expression by Nrf2 [134,152]. Curcumin provides medicinal benefits against diabetes, multiple sclerosis, Alzheimer’s, HIV disease, septic shock, cardiovascular diseases, lung fibrosis, arthritis and inflammatory bowel disease [114]. Curcumin may be a potential anticancer agent for both chemo-preventive and chemotherapeutic purposes [153]. It can effectively inhibit almost every stage of carcinogenesis [114]. Aqueous extract from Turmeric *Curcuma longa* shows antidepressant activity in mice following oral administration connected with inhibition of brain monoamine oxidase A (MAO) [154]. Our study revealed for the first time that curcumin may reduce prooxidative properties of antipsychotic Ziprasidone and protect against damage to plasma lipids and their peroxidation. Further research is needed to establish the interaction and action mechanism of curcumin and Ziprasidone.

## 4. Material and Methods

### 4.1. Material

Ziprasidone (active substance) was obtained from Pfizer Inc. (66 Hudson Blvd East, New York, NY 10001, USA) and *Curcuma longa* (Turmeric; Curcumin) from Sigma Aldrich (Sigma-Aldrich-Poland, Szelągowska 30, 61-626 Poznań, Poland). All the other reagents were of analytical grade and were provided by commercial suppliers. Stock solutions of Ziprasidone and curcumin were made in dimethyl-sulfoxide (DMSO). Ziprasidone was used in vitro at final concentrations corresponding to the doses used for the treatment of schizophrenia [96].

The sample size employed in this study can be considered sufficient for its stated objectives. As the work was conducted in vitro using human plasma, the primary requirement is not large-scale population representativeness but rather reproducibility and control of experimental conditions. The authors addressed this by including multiple biological donors and performing repeated measurements, which strengthens the reliability of the findings. Furthermore, the use of several concentrations of both curcumin and Ziprasidone, combined with time points aligned to pharmacokinetic considerations, provides a robust framework for assessing dose–response relationships. Taken together, these design features ensure that the sample is adequate to support the preliminary conclusions, while appropriately acknowledging that further validation in cellular and in vivo models will be necessary before translational claims can be made. Although the in vitro plasma model provided valuable preliminary data, it does not fully capture the complexity of in vivo physiology. Therefore, this limitation is acknowledged, and future studies should incorporate more physiologically relevant systems, such as cellular or animal models, to enhance translational applicability [155,156,157].

### 4.2. Inclusion Criteria of Healthy Controls

Blood samples were taken from 10 healthy controls aged between 24 and 26 years without psychiatric, neurological, or somatic disorders and without history of head injuries, allergies, and lipid or carbohydrate metabolism disorders, untreated with drugs. Healthy subjects did not use addictive substances, antioxidant supplementation (plants, pharmaceuticals) and polyunsaturated fatty acids. They lived in similar socioeconomic conditions, and their diet was balanced (meat and vegetables). Their body mass index (BMI) was in the normal range (18.5–24.9). Subjects with significant medical illnesses were excluded. There were no smokers. The Mini International Neuropsychiatric Interview (MINI) [158], neurological and somatic examinations, and total cholesterol, LDL, HDL, triglycerides, and glucose concentrations were performed. All healthy controls included in the study had been informed about the aims of the study and methods implemented and they expressed their written informed consent for participation in this study. The study was approved by the Committee for Research on Human Subjects of the Medical University of Łódź, Poland (number KE/719/09) and conducted according to the ethical obligations of the Declaration of Helsinki.

### 4.3. Isolation and Incubation of Plasma with Ziprasidone and Curcumin

Blood was collected separately into ACD solution (citric acid/citrate/dextrose; 5:1 *v*/*v*) and centrifuged (3000× *g*, 15 min.) to obtain plasma. Plasma was preincubated (5 min., 37 °C) with curcumin added in final concentrations of 5 µg/mL, 12.5 µg/mL, 25 µg/mL, 50 µg/mL. Ziprasidone solutions were added to the samples of plasma (0.5 mL) to obtain the final concentration of 40 ng/mL; 139 ng/mL; and 250 ng/mL. For each experiment, the control samples (without the drug) were prepared. The plasma samples with Ziprasidone in final concentrations and the control samples were incubated for 1 and 24 h at 37 °C. To measure the effects of curcumin, the samples of plasma were preincubated for 5 min. at 37 °C with curcumin solutions in 0.001% DMSO (final concentrations of curcumin: 5, 12.5, 25, 50 µg/mL) and then Ziprasidone (final concentrations: 40 ng/mL, 139 ng/mL; 250 ng/mL) was added. After 1 or 24 h incubation (37 °C) the level of TBARS was measured. The experiments were performed simultaneously, with 200 determinations, in duplicate, i.e., n = 800 determinations (Figure 5).

### 4.4. Estimation of Thiobarbituric Acid-Reactive Substances (TBARS) in Plasma

Samples of plasma after 1- or 24 h incubation with both Ziprasidone and curcumin or with Ziprasidone or curcumin alone were stopped by cooling the samples in an ice bath. Samples of plasma were transferred to an equal volume of 20% (*v/v*) cold trichloroacetic acid in 0.6 M HCL and centrifuged at 1200× *g* for 15 min. One volume of clear supernatant was mixed with 0.2 volume of 0.12 M thiobarbituric acid in 0.26 M Tris, pH 7.0, and immersed in a boiling water bath for 15 min. The absorbance was measured in the SEMCO spectrophotometer at 535 nm, according to the Rice-Evans method modified by Wachowicz and Kustroń (1992) [159]. The TBARS level expressed in μmol/L was calculated based on the absorbance value, using the molar extinction coefficient for TBARS (ε = 1.56 × 10^5^ M^−1^ × cm^−1^). All estimations were performed twice, including control samples, where spontaneous lipid peroxidation, without the influence of the drug on plasma, was measured. A paired Student’s *t*-test (related dependent samples) was used for calculations. The effect of curcumin (4 concentrations listed) on plasma TBARS concentrations was assessed, compared to control samples without curcumin, and the effect of curcumin on TBARS concentrations after its prior incubation with Ziprasidone was examined. In the experiments with curcumin and Ziprasidone, 200 analyses were performed for one incubation time and 200 for the other.

### 4.5. Statistical Analysis

Statistical analysis was performed using Statistica v. 13. We used the Pearson Chi^2^ test with Yates’ correction. In the first stage, we determined the correctness of the distribution of results in relation to the normal distribution using the Shapiro–Wilk test. In the case of analysis of groups with a distribution close to normal (*p* > 0.05), they were analyzed with Student’s *t*-test. Analysis of a larger number of sets with a normal distribution was performed using the ANOVA test. If any of the groups did not have a normal distribution (*p* < 0.05), they were analyzed using nonparametric, rank tests (Mann–Whitney U-test for comparison of two sets) or the Kruskal–Wallis test (analysis of a larger number of groups). To analyze the correlation of parameters, we used the Spearman rank correlation test. When verifying all analyzes we used the significance coefficient at the level of α = 0.05, thanks to which the variables were considered statistically significant at *p* < 0.05 [160,161]. All the values in this study were expressed as mean ± standard error of mean (SEM). In order to eliminate uncertain data, Grubbs’ test was performed. The statistical analysis of the difference between the control plasma (without drug) and plasma treated with Ziprasidone alone, curcumin alone or both Ziprasidone and curcumin was performed with a paired Student’s *t*-test.

## 5. Conclusions

Antioxidant properties of curcumin: in vitro studies have shown that curcumin reduces lipid peroxidation in human plasma induced by Ziprasidone. Protection against oxidative stress: the antioxidant activity of curcumin may play a role in limiting oxidative damage, which is relevant to the pathophysiology of schizophrenia and other neuropsychiatric disorders. Potential adjunctive therapeutic effect: curcumin may serve as a potential supplement in the treatment of patients receiving antipsychotic medications, helping to mitigate adverse effects associated with oxidative stress. Need for further research: additional in vivo and clinical studies are required to confirm the efficacy and safety of curcumin as an adjunct in antipsychotic therapy. Translational relevance: findings suggest that curcumin may be applicable not only in experimental models but also in clinical contexts, such as the treatment of schizophrenia, where oxidative stress and lipid damage are key components of pathogenesis.

Our in vitro findings demonstrate that curcumin possesses antioxidant properties, effectively reducing Ziprasidone-induced lipid peroxidation in human plasma. By limiting oxidative damage, curcumin may contribute to the protection against mechanisms relevant to the pathophysiology of schizophrenia and other neuropsychiatric disorders. These results also suggest a potential adjunctive therapeutic role for curcumin in patients receiving antipsychotic medications, where it could help mitigate adverse effects associated with oxidative stress. Nevertheless, further in vivo and clinical studies are required to confirm its efficacy and safety in this context. Importantly, the translational relevance of our findings indicates that curcumin may be applicable not only in experimental models but also in clinical practice, particularly in conditions such as schizophrenia, where oxidative stress and lipid damage are central to disease progression.

## Figures and Tables

**Figure 1 ijms-26-10430-f001:**
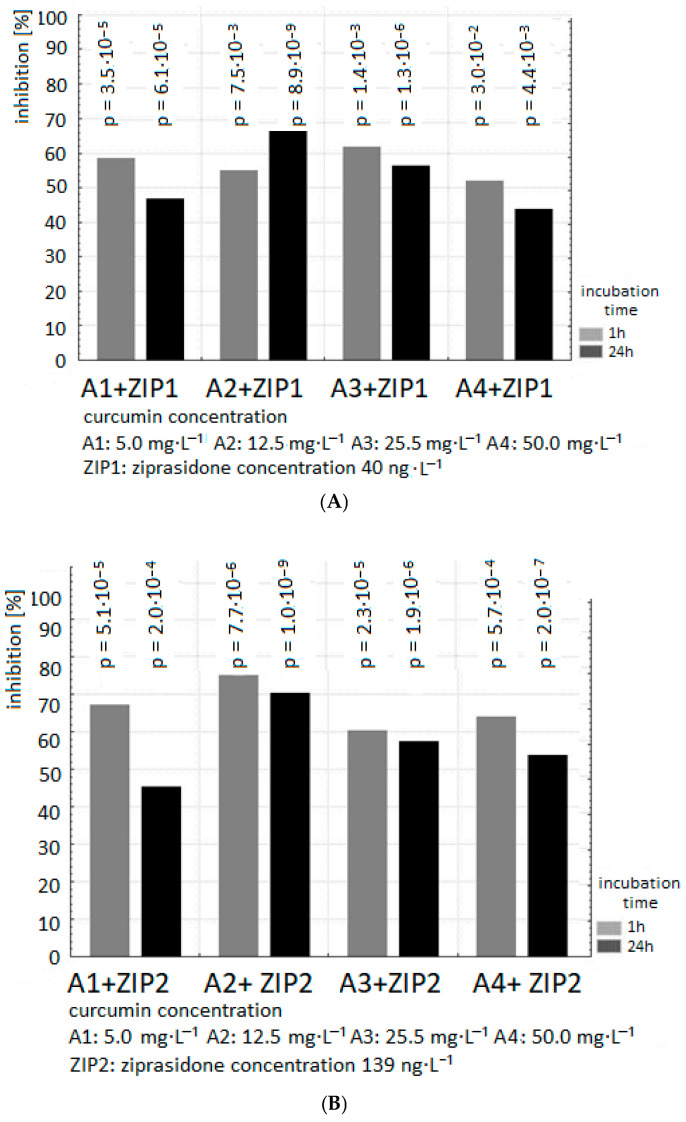
Effects of curcumin (5, 12.5, 25, 50 µg/mL) on the level of thiobarbituric acid-reactive substances (TBARS) in plasma incubated with ziprasidone for 1 h and 24 h. (**A**) 40 ng/mL ziprasidone, (**B**) 139 ng/mL ziprasidone, (**C**) 250 ng/mL ziprasidone. The TBARS level in control plasma (without ziprasidone) was 1.185 ± 0.04 µmol/L at 1 h and 1.56 ± 0.07 µmol/L at 24 h, and was set as 100%. Data are presented as a percentage of the control. Values are mean ± SD (n = 3).

**Figure 2 ijms-26-10430-f002:**
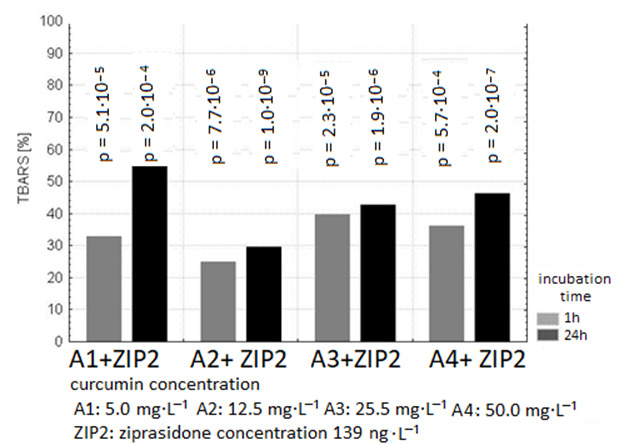
The effect of curcumin (concentration of curcumin: A1: 5 µg/mL; A2: 12.5 µg/mL; A3: 25 µg/mL; A4: 50 µg/mL) on TBARS level in plasma incubated with Ziprasidone at the concentration of 139 ng/mL for 1 h and 24 h. The level of TBARS in control plasma without Ziprasidone (1 h: 1.185 ± 0.04 µmol/L; 24 h: 1.56 ± 0.07 µmol/L) was expressed as 100%. The percentage values of the results are presented relative to the control value set at 100%.

**Figure 3 ijms-26-10430-f003:**
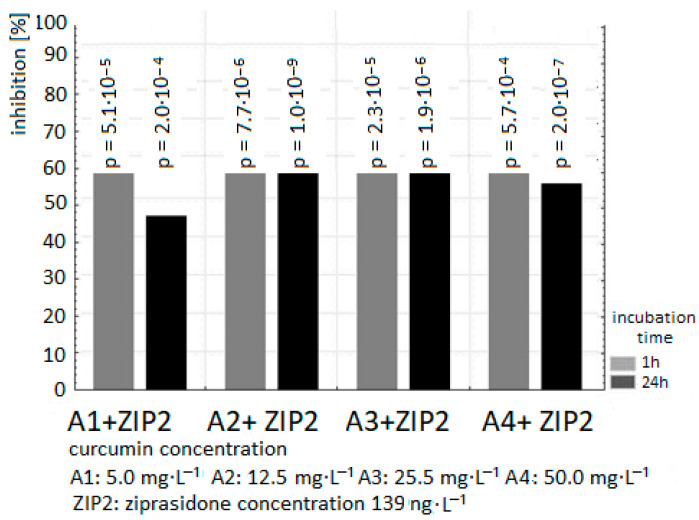
The effect of curcumin (concentration of curcumin: A1: 5 µg/mL; A2: 12.5 µg/mL; A3: 25 µg/mL; A4: 50 µg/mL) on the inhibition level in plasma incubated with Ziprasidone at the concentration of 139 ng/mL for 1 h and 24 h. The level of TBARS in control plasma without Ziprasidone (1 h: 1.185 ± 0.04 µmol/L; 24 h: 1.56 ± 0.07 µmol/L) was expressed as 100%. The percentage values of the results are presented relative to the control value set at 100%.

**Figure 4 ijms-26-10430-f004:**
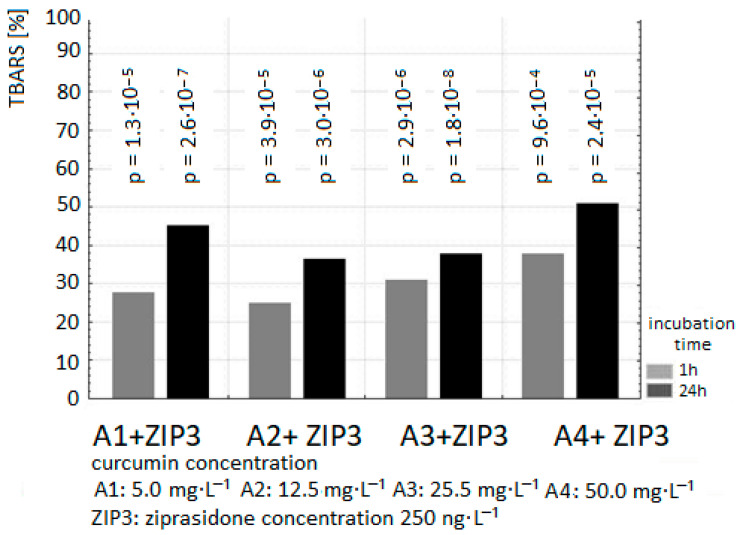
The effect of curcumin (concentration of curcumin: A1: 5 µg/mL; A2: 12.5 µg/mL; A3: 25 µg/mL; A4: 50 µg/mL) on TBARS level in plasma incubated with Ziprasidone at the concentration of 250 ng/mL for 1 h and 24 h. The level of TBARS in control plasma without Ziprasidone (1 h: 1.185 ± 0.04 µmol/L; 24 h: 1.56 ± 0.07 µmoL/L) was expressed as 100%. The percentage values of the results are presented relative to the control value set at 100%.

**Figure 5 ijms-26-10430-f005:**
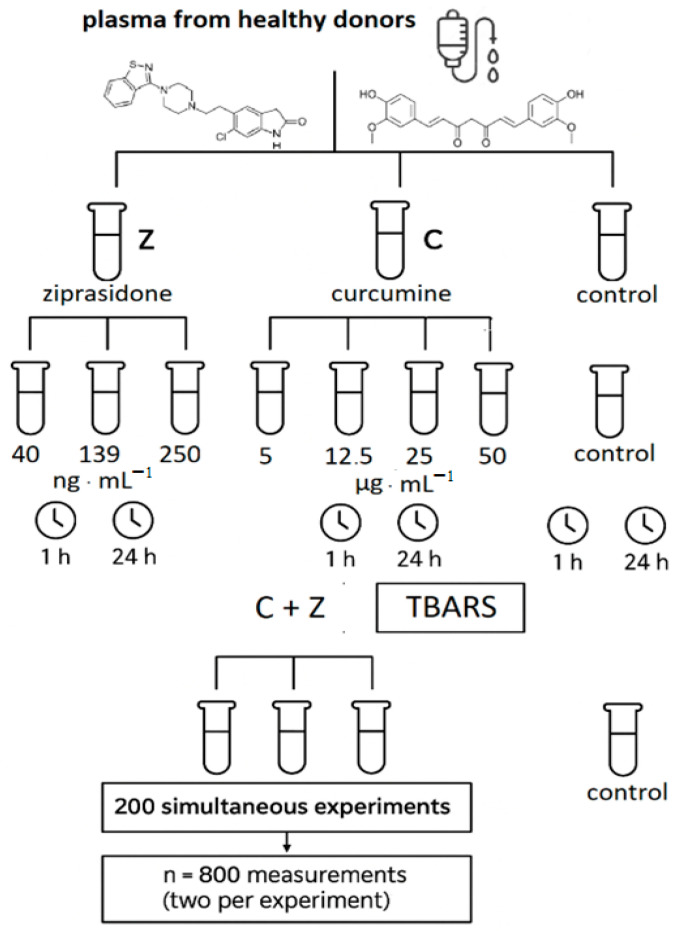
Schematic diagram of the analysis procedure used.

**Table 1 ijms-26-10430-t001:** Significant differences between curcumin and Ziprasidone concentration; time of incubation: 24 h. SEM—standard error of mean.

Concentration		Mean ± SEM	*p*	Inhibition %
Curcumin (A)	Ziprasidone (ZIP)			
A1: 5 µg/mL	and ZIP1: 40 ng/mL	0.831 ± 0.118	6.1 × 10^−5^	46.7
A2: 12.5 µg/mL	and ZIP1: 40 ng/mL	0.468 ± 0.069	8.9 × 10^−9^	70.0
A3: 25 µg/mL	and ZIP1: 40 ng/mL	0.680 ± 0.076	1.3 × 10^−6^	56.4
A4: 50 µg/mL	and ZIP1: 40 ng/mL	0.879 ± 0.179	4.4 × 10^−3^	43.6
A1: 5 µg/mL	and ZIP2: 139 ng/mL	0.852 ± 0.122	2.0 × 10^−4^	45.4
A2: 12.5 µg/mL	and ZIP2: 139 ng/mL	0.462 ± 0.052	1.0 × 10^−9^	70.4
A3: 25 µg/mL	and ZIP2: 139 ng/mL	0.665 ± 0.084	1.9 × 10^−6^	57.4
A4: 50 µg/mL	and ZIP2: 139 ng/mL	0.722 ± 0.049	2.0 × 10^−7^	53.8
A1: 5 µg/mL	and ZIP3: 250 ng/mL	0.706 ± 0.082	2.6 × 10^−7^	54.7
A2: 12.5 µg/mL	and ZIP3: 250 ng/mL	0.570 ± 0.092	3.0 × 10^−6^	63.5
A3: 25 µg/mL	and ZIP3: 250 ng/mL	0.590 ± 0.061	1.8 × 10^−8^	62.2
A4: 50 µg/mL	and ZIP3: 250 ng/mL	0.793 ± 0.084	2.4 × 10^−5^	49.2

**Table 2 ijms-26-10430-t002:** Significant differences between thiobarbituric acid-reactive substances’ (TBARSs’) activity; time of incubation: 1 h and 24 h. K—the reference value, i.e., the TBARS activity expressed as a percentage of the control. K = 100.0 for the control means that the TBARS value in this group is taken as the reference point (100%). For other conditions (e.g., A1/L1, A2/L2, etc.), K shows what proportion of TBARS activity a given mean value represents compared to the control. SEM—standard error of mean.

TBARS 1 h	Mean	SEM	*p*	K	Inhibition %
Control	1.185	0.038		100.0	
A1 5/L1 40 ng/mL	0.493	0.112	0.003458	41.6	58.4
A2 12.5/L1 40 ng/mL	0.533	0.119	0.007488	45.0	55.0
A3 25/L1 40 ng/mL	0.454	0.098	0.001407	38.3	61.7
A4 50/L1 40 ng/mL	0.570	0.188	0.029623	48.1	51.9
A1 5/L2 139 ng/mL	0.389	0.055	5.13 × 10^−5^	32.8	67.2
A2 12.5/L2 139 ng/mL	0.294	0.035	7.71 × 10^−6^	24.8	75.2
A3 25/L2 139 ng/mL	0.469	0.041	2.32 × 10^−5^	39.6	60.4
A4 50/L2 139 ng/mL	0.427	0.099	0.000572	36.0	64.0
A1 5/L3 250 ng/mL	0.327	0.053	1.32 × 10^−5^	27.6	72.4
A2 12.5/L3 250 ng/mL	0.294	0.058	3.94 × 10^−5^	24.8	75.2
A3 25/L3 250 ng/mL	0.366	0.034	2.88 × 10^−6^	30.8	69.2
A4 50/L3 250 ng/mL	0.446	0.121	0.000959	37.7	62.3
TBARS 24 h	mean	SEM	*p*	K	inhibition %
Control	1.560	0.074		100.0	
A1 5/L1 40 ng/mL	0.831	0.118	6.15 × 10^−5^	53.3	46.7
A2 12.5/L1 40 ng/mL	0.468	0.069	8.87 × 10^−9^	30.0	70.0
A3 25/L1 40 ng/mL	0.680	0.076	1.26 × 10^−6^	43.6	56.4
A4 50/L1 40 ng/mL	0.879	0.179	0.004428	56.4	43.6
A1 5/L2 139 ng/mL	0.852	0.122	0.000199	54.6	45.4
A2 12.5/L2 139 ng/mL	0.462	0.052	1.03 × 10^−9^	29.6	70.4
A3 25/L2 139 ng/mL	0.665	0.084	1.92 × 10^−6^	42.6	57.4
A4 50/L2 139 ng/mL	0.722	0.049	1.99 × 10^−7^	46.2	53.8
A1 5/L3 250 ng/mL	0.706	0.082	2.6 × 10^−7^	45.3	54.7
A2 12.5/L3 250 ng/mL	0.570	0.092	3.01 × 10^−6^	36.5	63.5
A3 25/L3 250 ng/mL	0.590	0.061	1.81 × 10^−8^	37.8	62.2
A4 50/L3 250 ng/mL	0.793	0.084	2.42 × 10^−5^	50.8	49.2

## Data Availability

The data presented in this study are available on request from the corresponding author.

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
