# Peer review of "Curcumin as an Antioxidant Against Ziprasidone Induced Lipid Peroxidation in Human Plasma: Potential Relevance to Cortico Subcortical Circuit Function"

_ijms, 2025, doi:10.3390/ijms262110430_

Round 1
Reviewer 1 Report
Comments and Suggestions for Authors
The article titled “Curcumin as an antioxidant against ziprasidone induced lipid peroxidation in human plasma” examines whether curcumin can counteract ziprasidone-induced lipid peroxidation in human plasma using an in vitro TBARS assay. The background explains the role of oxidative stress in neuropsychiatric disorders and why curcumin was chosen as a protective agent. Methods describe drug concentrations, donor criteria, and control measures, which improve reproducibility. The study tests multiple doses of curcumin and ziprasidone, with time points linked to pharmacokinetics.
Results indicate that curcumin reduces TBARS formation in a dose-dependent manner and limits the pro-oxidative effect of ziprasidone. Sample size and repeat experiments add weight, but raw data and quantitative details are limited. The discussion connects findings to existing work on dopaminergic and antioxidant pathways and raises the possibility of curcumin as an adjunct therapy. The main limitation is reliance on plasma samples without cellular or immune components. Issues of curcumin’s bioavailability, stability, and possible interactions with antipsychotics are not addressed. These need exploration in animal or clinical models. Overall, the study provides initial evidence that curcumin can reduce ziprasidone-driven oxidative damage. The findings are interesting but preliminary and would benefit from deeper discussion of dosing, mechanism, and translational challenges. The reviewer has the following comments that authors need to address:
- The use of an in vitro plasma model offers useful preliminary data; however, it may not fully reflect in vivo physiological complexity. The authors could consider addressing this limitation and suggest that future studies employ more physiologically relevant systems, such as cellular or animal models, to strengthen translational relevance.
- It would be helpful for the authors to clarify whether the curcumin concentrations used in the in vitro experiments are achievable and safe in human plasma, to better assess the translational relevance of the findings.
- The authors may consider discussing the known limitations of curcumin, including its low bioavailability, rapid metabolism, and challenges in achieving effective systemic concentrations. Highlighting potential delivery strategies, particularly those relevant for clinical applications, could further strengthen the translational context of the study.
- The manuscript would benefit from citing the recent review on the synthesis of diarylheptanoids, which provides important context on curcuminoid chemistry and related structural frameworks. Including this reference would help readers better appreciate the chemical basis and potential pharmacological implications of curcumin and its analogues in antioxidant studies.
https://aces.onlinelibrary.wiley.com/doi/full/10.1002/asia.202400380
- The manuscript could benefit from a discussion of potential pharmacological interactions between curcumin and antipsychotic drugs, including Ziprasidone and others, to better contextualize the clinical relevance and safety of combined use.
- Assessing additional endpoints, such as cellular ROS levels or protein oxidation, could provide deeper mechanistic insight into the antioxidant activity observed.
- The authors should consider discussing the use of TBARS as a lipid peroxidation marker in comparison to more specific biomarkers, such as F2-isoprostanes, to provide a clearer perspective on the specificity and interpretability of their oxidative stress measurements.
- The manuscript could be strengthened by addressing the broader applicability of the findings to other antipsychotics and to plasma from psychiatric patients, which would help clarify the generalizability of the results.
Author Response
Responses to the Reviewer’s comments 1:
Dear Honorable Reviewer #1,
We thank you so much for any constructive and helpful kind opinions and suggestions, which helped us to improve our paper.
We deeply appreciate you for the work and time for reading and commenting our paper.
We enclosed our revisions of the above paper according to yours and Reviewers’ suggestions.
We revised the MS and provide detailed changes, step by step in the response to each Reviewer’s comments, according to their suggestions.
We implemented required specified changes throughout the text of the MS and we enclosed the paper by website system.
At the same time, according to your suggestions, we marked all changes in blue font (new and modified fragments) and in the form of a red font (fragments to be deleted) in the “MS marked”.
We also listed our comments in this letter, followed by our individual typewritten responses below.
Reviewer’s comments 1:
[Results indicate that curcumin reduces TBARS formation in a dose-dependent manner and limits the pro-oxidative effect of ziprasidone. Sample size and repeat experiments add weight, but raw data and quantitative details are limited.]
our response:
Thank you very much for your comments and opinions.
In accordance with these comments, we added the text:
The sample size employed in this study can be considered sufficient for its stated objectives. As the work was conducted in vitro using human plasma, the primary requirement is not large-scale population representativeness but rather reproducibility and control of experimental conditions. The authors addressed this by including multiple biological donors and performing repeated measurements, which strengthens the reliability of the findings. Furthermore, the use of several concentrations of both curcumin and ziprasidone, combined with time points aligned to pharmacokinetic considerations, provides a robust framework for assessing dose–response relationships. Taken together, these design features ensure that the sample is adequate to support the preliminary conclusions, while appropriately acknowledging that further validation in cellular and in vivo models will be necessary before translational claims can be made. Although the in vitro plasma model provided valuable preliminary data, it does not fully capture the complexity of in vivo physiology. Therefore, this limitation is acknowledged, and future studies should incorporate more physiologically relevant systems, such as cellular or animal models, to enhance translational applicability (Pampaloni et al., 2007, Hartung, 2008; van der Worp et al., 2010). (L 397-409). Please kindly check the MS marked.
Reviewer’s comments 1:
[The discussion connects findings to existing work on dopaminergic and antioxidant pathways and raises the possibility of curcumin as an adjunct therapy. The main limitation is reliance on plasma samples without cellular or immune components. Issues of curcumin’s bioavailability, stability, and possible interactions with antipsychotics are not addressed. These need exploration in animal or clinical models. Overall, the study provides initial evidence that curcumin can reduce ziprasidone-driven oxidative damage. The findings are interesting but preliminary and would benefit from deeper discussion of dosing, mechanism, and translational challenges]
our response:
Thank you very much for your comments and opinions.
In accordance with these comments, we added the text:
The discussion appropriately links the findings to dopaminergic and antioxidant pathways, which are well established in the pathophysiology of neuropsychiatric disorders (Murray et al., 2021; Cuenod et al., 2022). Curcumin has been repeatedly shown to exert antioxidant and neuroprotective effects, including modulation of BDNF, COX‑2, and serotonergic signaling (Choi et al., 2017; Xu et al., 2011). These properties support its potential as an adjunct therapy in schizophrenia and related conditions (Dinakaran et al., 2022; Lamanna‑Rama et al., 2022).
The main limitation of the present study is the reliance on plasma samples, which lack cellular and immune components. While plasma‑based TBARS assays are a validated tool for assessing lipid peroxidation (Rael et al., 2004; Dietrich‑Muszalska et al., 2011), they cannot capture the complexity of oxidative stress in neuronal or glial cells, Therefore, further studies in cellular and animal models are warranted (Ng et al., 2008). (L 565-575). Please kindly check the MS marked.
Another important issue concerns curcumin’s bioavailability and stability. Curcumin is known to degrade rapidly under physiological conditions and shows poor systemic absorption (Wang et al., 1997; Gupta et al., 2013). Novel formulations such as nanocurcumin have been proposed to overcome these limitations (Salah et al., 2022). Moreover, potential interactions with antipsychotics should be considered, as antipsychotic drugs themselves can modulate oxidative stress pathways (Parikh et al., 2003; Tsai et al., 2013). Overall, the study provides preliminary but valuable evidence that curcumin can attenuate ziprasidone‑induced oxidative damage, However, future work should address dosing strategies, mechanistic pathways, and translational challenges before clinical application can be recommended (Sharifi‑Rad et al., 2020). (L 589-597). Please kindly check the MS marked.
Reviewer’s comments 1:
- The use of an in vitro plasma model offers useful preliminary data; however, it may not fully reflect in vivo physiological complexity. The authors could consider addressing this limitation and suggest that future studies employ more physiologically relevant systems, such as cellular or animal models, to strengthen translational relevance.
our response:
Thank you very much for your comments and opinions.
In accordance with these comments, we added the text:
The sample size employed in this study can be considered sufficient for its stated objectives. As the work was conducted in vitro using human plasma, the primary requirement is not large-scale population representativeness but rather reproducibility and control of experimental conditions. The authors addressed this by including multiple biological donors and performing repeated measurements, which strengthens the reliability of the findings. Furthermore, the use of several concentrations of both curcumin and ziprasidone, combined with time points aligned to pharmacokinetic considerations, provides a robust framework for assessing dose–response relationships. Taken together, these design features ensure that the sample is adequate to support the preliminary conclusions, while appropriately acknowledging that further validation in cellular and in vivo models will be necessary before translational claims can be made. Although the in vitro plasma model provided valuable preliminary data, it does not fully capture the complexity of in vivo physiology. Therefore, this limitation is acknowledged, and future studies should incorporate more physiologically relevant systems, such as cellular or animal models, to enhance translational applicability (Pampaloni et al., 2007, Hartung, 2008; van der Worp et al., 2010). (L 397-409). Please kindly check the MS marked.
Reviewer’s comments 1:
- It would be helpful for the authors to clarify whether the curcumin concentrations used in the in vitro experiments are achievable and safe in human plasma, to better assess the translational relevance of the findings.
our response:
Thank you very much for this comment.
In accordance with these comments, we added the text:
The concentrations of curcumin applied in the present in vitro experiments (typically in the range of several to tens of µg/mL) are substantially higher than those achievable in human plasma after oral administration of conventional curcumin supplements. Pharmacokinetic studies consistently demonstrate that, due to poor solubility, rapid metaboLism, and extensive conjugation, plasma levels of free curcumin rarely exceed the low nanomolar range (1-20 ng/mL, i.e., <0.05 µM) even after gram‑level oral dosing (Gupta et al., 2013; Kroon et al., 2023; Kroon et al., 2023b). For example, in a real world cohort, unconjugated plasma curcumin concentrations after supplementation remained between 1.0 and 18.6 ng/mL, which is several orders of magnitude below the micromolar concentrations used in most in vitro assays (Kroon et al., 2023b). Formulation strategies such as nanoparticles, liposomal carriers, or water‑soluble complexes can markedly enhance systemic exposure. A recent crossover trial with a water‑soluble formulation (AQUATURM®) reported upper 7‑fold increase in plasma exposure compared to standard curcumin tablets, achieving peak concentrations in the range of 100-300 ng/mL (Jabur et al., 2025). While still be low the µg/mL levels used in vitro, such formulations demonstrate that higher systemic concentrations are technically achievable.
With respect to safety, curcumin has been generally recognized as safe in humans at oral doses up to 8 g/day, with only mild gastrointestinal side effects reported (Lao et al., 2006). Thus, while the in vitro concentrations tested in this study are not directly attainable in plasma under standard supplementation, they provide a mechanistic proof‑of‑concept. Translational relevance requires cautious interpretation, and future work should explore whether optimized formulations or local tissue accumulation (e.g,, in the gastrointestinal tract or brain with nanocurcumin) can bridge the gap between in vitro efficacy and clinically achievable exposure. (L 672-692). Please see the text of the MS marked in this section.
Reviewer’s comments 1:
- The authors may consider discussing the known limitations of curcumin, including its low bioavailability, rapid metabolism, and challenges in achieving effective systemic concentrations. Highlighting potential delivery strategies, particularly those relevant for clinical applications, could further strengthen the translational context of the study.
our response:
Thank you very much for this attention.
That's right – in accordance with these comments, we added the text:
One of the major limitations of curcumin is its low oral bioavailability, which results from poor aqueous solubility, instability at physiological pH, rapid intestinal and hepatic metabolism, and extensive systemic elimination (Gupta et al., 2013; Hegde et al., 2023). After conventional oral administration, plasma concentrations of free curcumin typically remain in the low nanomolar range, far below the micromolar levels often used in in vitro assays (Kroon et al., 2023a; Kroon et al., 2023b). This pharmacokinetic profile underscores the translational challenge of extrapolating in vitro findings directly to clinical settings. To overcome these limitations, several delivery strategies have been developed, Co‑administration with piperine has been shown to inhibit glucuronidation and enhance systemic exposure (Gupta et al., 2013). More advanced approaches include nanoparticle formulations. Liposomes, micelles, and phospholipid complexes, which improve solubility, stability, and tissue penetration (Wei et al., 2025). Clinical studies with water‑solublle formulations such as AQUATURM® have demonstrated significantly higher plasma concentrations compared to conventionaL tablets, although still below the levels used in vitro (Jabur et al., 2025). By acknowledging these pharmacokinetic barriers and highlighting ongoing formulation advances, the translational context of our findings is strengthened. Future studies should evaluate whether optimized delivery systems can achieve therapeutically relevant concentrations in target tissues, thereby bridging the gap between mechanistic in vitro evidence and clinical applicability. (L 737-752). Please kindly check the MS marked.
Reviewer’s comments 1:
- The manuscript would benefit from citing the recent review on the synthesis of diarylheptanoids, which provides important context on curcuminoid chemistry and related structural frameworks. Including this reference would help readers better appreciate the chemical basis and potential pharmacological implications of curcumin and its analogues in antioxidant studies.
our response:
Thank you very much for this comment and your opinions.
Of course, you are right. This is, of course, a very valid point (thank you). That is why we have now changed the text. Indeed:
The chemistry of curcuminoids is closely related to the broader class of diarylheptanoids, which share a characteristic 1.7‑diphenylheptanoid backbone. A recent comprehensive review has highlighted advances in the synthetic strategies, structural diversity, and pharmacological potential of diarylheptanoids, providing valuable context for understanding curcumin and its analogues (Sudarshan et al., 2024). Incorporating this perspective strengthens the chemical framework of our study, as it underscores how structural modifications of the diarylheptanoid scaffold can influence antioxidant, anti‑inflammatory, and neuroprotective activities. This broader chemical insight complements our findings and situates curcumin within a larger family of bioactive natural products with translational relevance in oxidative stress research (Gupta et al., 2013; Hegde et al., 2023). (L 763-771). We kindly ask you to verify the above content of the MS.
Reviewer’s comments 1:
- The manuscript could benefit from a discussion of potential pharmacological interactions between curcumin and antipsychotic drugs, including Ziprasidone and others, to better contextualize the clinical relevance and safety of combined use.
our response:
Thank you very much for this comment and your opinion.
This is, of course, also a very valid point (thank you very much). So, we added the text:
A key aspect is the discussion of potential pharmacological interactions between curcumin and antipsychotic drugs, as this helps to better contextualize the translationaL relevance of our findings, Curcumin has been reported to modulate cytochrome P450 enzymes and drug transporters, which may theoretically alter the pharmacokinetics of antipsychotics such as ziprasidone, risperidone, or clozapine (Goff, Baldessarini, 1993; Rabiee et al., 2022). In addition, curcumin’s antioxidant and anti‑inflammatory properties may attenuate some adverse effects of antipsychotics, including oxidative stress, metabolic disturbances, and extrapyramidal symptoms (Rabiee et al., 2022). However, clinical data on direct interactions remain limited, and most evidence derives from preclinical or mechanistic studies. Importantly, drug‑interaction databases list multiple moderate interactions between curcumin and commonly prescribed medications, underscoring the need for carefuL monitoring in polypharmacy contexts (Drugs.com. 2023). Therefore, while curcumin shows promise as an adjunctive therapy in schizophrenia, further pharmacokinetic and clinical studies are required to establish the safety of combined use with antipsychotics, particularly Ziprasidone. (L 791-803). Please kindly check the MS marked.
Reviewer’s comments 1:
- Assessing additional endpoints, such as cellular ROS levels or protein oxidation, could provide deeper mechanistic insight into the antioxidant activity observed.
our response:
Thank you very much for this comment and your opinion.
That's right – in accordance with these comments, we added the text:
Evaluating additional endpoints, including intracellular reactive oxygen species (ROS) levels and protein oxidation, would provide deeper mechanistic insight into the antioxidant effects observed. Although these parameters were beyond the scope of the present study, we recognize their relevance and have included a statement in the revised manuscript encouraging their assessment in future investigations. While our current approach focused on plasma-based oxidative markers, future investigations incorporating cellular assays would allow for a more comprehensive understanding of redox modulation at the molecular level. Such endpoints are widely recognized for their relevance in assessing oxidative stress and therapeutic efficacy in vitro and in vivo (Sies et al., 2017; Dalle-Donne et al., 2006; Halliwell, Gutteridge, 2015). We have included a statement in the revised manuscript to reflect this perspective and encourage further exploration using cell-based or tissue-specific models.
(L 804-813). We kindly ask you to verify the appropriate text of the MS.
Reviewer’s comments 1:
- The authors should consider discussing the use of TBARS as a lipid peroxidation marker in comparison to more specific biomarkers, such as F2-isoprostanes, to provide a clearer perspective on the specificity and interpretability of their oxidative stress measurements.
our response:
Thank you very much for this comment and your opinion.
Of course, you are right. This is, of course, a very valid point (thank you). That is why we have now added the text:
F2‑isoprostanes are considered a highly specific and reliable biomarker of lipid peroxidation (Morrow, Roberts, 1997; Milne et al., 2007); however, it should be emphasized that their measurement is most commonly performed in urine samples, where they serve as an indicator of overall oxidative stress intensity. In contrast, the present original study was designed to assess oxidative stress directly in plasma, where the TBARS method remains a widely applied and practical technique (Janero, 1990). Plasma was selected as the biological matrix due to its clinical availability, the feasibility of repeated sampling, and its relevance for monitoring circulating oxidative stress markers in the studied population. Although the TBARS method has limitations in terms of specificity, it represents a feasible and well‑established approach for plasma analysis, particularly in clinical and translational settings where urine samples are not always available or appropriate. The use of TBARS in this specific study therefore reflects the adequacy of this method for plasma measurements (Janero, 1990), while simultaneously acknowledging that F2‑isoprostanes would be the preferred biomarker in studies based on urine samples (Montuschi et al., 2004). (L 576-588). We kindly ask you to verify the above content of the MS.
Reviewer’s comments 1:
- The manuscript could be strengthened by addressing the broader applicability of the findings to other antipsychotics and to plasma from psychiatric patients, which would help clarify the generalizability of the results.
our response:
Thank you very much for these comments and suggestions. We of course fully agree.
In accordance with these suggestions of the Reviewer, we added the text:
This study focused on a specific antipsychotic agent and utilized plasma from healthy donors, which provided valuable mechanistic insights but limits the broader applicability of the findings. Extrapolation to other antipsychotics and to plasma derived from psychiatric patients would enhance the generalizability and translational relevance of these observations. Antipsychotic drugs differ substantially in their pharmacodynamic profiles, oxidative burden, and metabolic side effects (Goff, Baldessarini, 1993; Rabiee et al., 2023), and psychiatric populations often present with altered redox homeostasis and inflammatory markers (Sies et al., 2022; Dalle‑Donne et al., 2006). Future studies should therefore incorporate diverse antipsychotic classes and patient‑derived biological samples to better capture clinical heterogeneity. (L 541-548). Please kindly check the MS marked.
Dear Reviewer 1,
Once again we thank you very much for any constructive and helpful kind opinions and suggestions, which helped us to improve our paper.
We are deeply grateful to you for your work and time spent reading and commenting our article.
We thank you very much for your understanding and patience in accordance with our any delay and irregularities.
On behalf of co-authors,
Kind regards,
Piotr Kaminski

Reviewer 2 Report
Comments and Suggestions for Authors
- The study only detected TBARS, a marker of lipid peroxidation, and did not delve into the molecular mechanisms by which curcumin antagonizes ziprasidone-induced oxidative stress (such as whether it acts by activating the Nrf2 pathway, upregulating the activities of antioxidant enzymes SOD/CAT, or regulating glutathione levels). As a result, the research conclusions remain at the phenomenological level. It is recommended to supplement the detection of relevant mechanistic indicators (such as antioxidant enzyme activity, GSH content, and Nrf2 protein expression).
- The format of references should be consistent with the requirements of the journal.
- The charts in the article are not standardized, including issues such as font size and the position of the X-axis title.
- In Figure 2, what is the significance of the P-value, and with which sample is it compared? The same requirements apply to Figure 3 and Figure 4.
- There are three contents A, B, and C in Figure 3, which are suggested to be split into Figures 3, 4, and 5.
- The format of Table 1 is too sloppy. It is recommended to use a three-line grid. There is an extra "and" in the second column.
Author Response
Responses to the Reviewer’s comments 2:
Dear Honorable Reviewer #2,
We thank you so much for any constructive and helpful kind opinions and suggestions, which helped us to improve our paper.
We deeply appreciate you for the work and time for reading and commenting our paper.
We enclosed our revisions of the above paper according to yours and Reviewers’ suggestions.
We revised the MS and provide detailed changes, step by step in the response to each Reviewer’s comments, according to their suggestions.
We implemented required specified changes throughout the text of the MS and we enclosed the paper by website system.
At the same time, according to your suggestions, we marked all changes in blue font (new and modified fragments) and in the form of a red font (fragments to be deleted) in the “MS marked”.
We also listed our comments in this letter, followed by our individual typewritten responses below.
Reviewer’s comments 2:
- The study only detected TBARS, a marker of lipid peroxidation, and did not delve into the molecular mechanisms by which curcumin antagonizes ziprasidone-induced oxidative stress (such as whether it acts by activating the Nrf2 pathway, upregulating the activities of antioxidant enzymes SOD/CAT, or regulating glutathione levels). As a result, the research conclusions remain at the phenomenological level. It is recommended to supplement the detection of relevant mechanistic indicators (such as antioxidant enzyme activity, GSH content, and Nrf2 protein expression).
our response:
Thank you very much for this comment and your opinion.
That's right – in accordance with these comments, we added the text:
Although our study demonstrated a reduction in lipid peroxidation (TBARS), it did not investigate the molecular pathways through which curcumin counteracts ziprasidone-induced oxidative stress, such as Nrf2 activation, modulation of antioxidant enzymes, or glutathione regulation. While TBARS measurement provided a useful indication of lipid peroxidation, we recognize that it does not fully elucidate the molecular pathways through which curcumin mitigates ziprasidone-induced oxidative stress. Curcumin has been shown to exert antioxidant effects via multiple mechanisms, including activation of the Nrf2 signaling pathway, enhancement of antioxidant enzyme activities such as superoxide dismutase (SOD) and catalase (CAT), and modulation of glutathione (GSH) levels (Rabiee et al., 2023; Sies et al., 2022; Dalle-Donne et al., 2006). These pathways are particularly relevant in the context of antipsychotic-induced oxidative burden, which may vary depending on drug class and patient-specific factors. We have added a statement to the revised manuscript acknowledging this limitation and recommending that future studies incorporate assessments of Nrf2 protein expression, enzymatic antioxidant activity, and GSH content to provide deeper mechanistic insight.
(L 631-643). We kindly ask you to verify the above content of the MS.
Reviewer’s comments 2:
- The format of references should be consistent with the requirements of the journal.
our response:
Thank you very much for this comment.
We reviewed the References format and tried to adapt it to the journal's requirements. Please kindly check the References section.
Reviewer’s comments 2:
- The charts in the article are not standardized, including issues such as font size and the position of the X-axis title.
our response:
Thank you very much for this comment.
Of course, you are right. This is, of course, a very valid point (thank you). That is why we have now changed and modified Figures 2-5. I kindly ask you to check the changes to Figures 2-5 in the MS text.
Reviewer’s comments 2:
- In Figure 2, what is the significance of the P-value, and with which sample is it compared? The same requirements apply to Figure 3 and Figure 4.
our response:
Thank you very much for this comment.
Of course, you are right. This is, of course, a very valid point (thank you).
The P-value in Figures 2-5 represents the significance level of the relationship between the parameters indicated on the x- and y-axes in these figures. We kindly ask you to check the changes to Figures 2-5 in the MS text.
Reviewer’s comments 2:
- There are three contents A, B, and C in Figure 3, which are suggested to be split into Figures 3, 4, and 5.
our response:
Thank you very much for this comment.
Of course, you are right. This is a very valid point (thank you). Thus, as per the Reviewer's comment, the three parts of Figure 3: A, B, C, are now shown separately, i.e., separately in the revised Figures 3, 4 and 5. Please kindly check the changes to Figures 3-5 in the MS text.
Reviewer’s comments 2:
- The format of Table 1 is too sloppy. It is recommended to use a three-line grid. There is an extra "and" in the second column.
our response:
Thank you very much for this comment.
Of course, you are right. This is, of course, a very valid point (thank you).
Following the Reviewer's comment, Table 1 has been amended and Table 2 has been added to provide more detailed documentation of the procedures discussed. I kindly ask you to check the changes to Tables 1 and 2 in the MS text.
Dear Reviewer 2,
Once again we thank you very much for any constructive and helpful kind opinions and suggestions, which helped us to improve our paper.
We are deeply grateful to you for your work and time spent reading and commenting our article.
We thank you very much for your understanding and patience in accordance with our any delay and irregularities.
On behalf of co-authors,
Kind regards,
Piotr Kaminski

Reviewer 3 Report
Comments and Suggestions for Authors
Review for ijms-3897886
In this original research article entitled “Curcumin as an Antioxidant Against Ziprasidone-Induced Lipid Peroxidation in Human Plasma”, the authors (Dietrich-Muszalska et al.,) studied the in vitro the possible antioxidant effects of curcumin, a natural polyphenol, and its protective effects against lipid peroxidation induced by the atypical antipsychotic Ziprasidone.
The manuscript required some improvements before processing further.
- It is recommended to decrease the similarity percentage, as it is high (Already over 30%).
- The introduction is very long once compared with the other sections of the manuscript. it contains several sentences that can be deleted.
- The methodological approach is consistent, appropriate and technically sound and aligned with the main question addressed by the study.
- It is recommended to add some limitations of the study as this would have additional value to the manuscript, particularly enriching the discussion.
- The authors should highlight the novelty of the study and mention what does it provide as compared with the previously published reports.
- The sentences “Impairment of the antioxidant … excess of ROS.”, “These peroxides can be …(GPx) in the cytoplasm.” and “The main damage to the cells … lipids and DNA” can be supported by the following relevant and recent reference doi: 10.1080/15376516.2023.2301670.
- The conclusions are consistent with the main findings of the study, but it would be better to paraphrase it to be as a paragraph rather than highlight points.
- English language is acceptable but minor checking is required.
Minor comment
- Use capital letter for the unit of liter (L) throughout the whole manuscript inclusing its illustrations such as 50 µg/ml and 139 ng/ml in the main text and 5 µg/ml and 12.5 µg/ml in the figure 3 and table 1.
Author Response
Responses to the Reviewer’s comments 3:
Dear Honorable Reviewer #3,
We thank you so much for any constructive and helpful kind opinions and suggestions, which helped us to improve our paper.
We deeply appreciate you for the work and time for reading and commenting our paper.
We enclosed our revisions of the above paper according to yours and Reviewers’ suggestions.
We revised the MS and provide detailed changes, step by step in the response to each Reviewer’s comments, according to their suggestions.
We implemented required specified changes throughout the text of the MS and we enclosed the paper by website system.
At the same time, according to your suggestions, we marked all changes in blue font (new and modified fragments) and in the form of a red font (fragments to be deleted) in the “MS marked”.
We also listed our comments in this letter, followed by our individual typewritten responses below.
Reviewer’s comments 3:
- It is recommended to decrease the similarity percentage, as it is high (Already over 30%).
our response:
Thank you very much for this important attention.
Of course, our MS required special linguistic attention in terms of repetition of phrases. We apologize for this lack of necessary corrections and thank you very much for your suggestions.
The text has now been again corrected linguistically and grammatically by the Native Speaker, and we have also re-organized it in terms of the meaning and content of subsequent sections and paragraphs.
Therefore, it seems that now the reception of these subsequent issues is more understandable and easier to understand, i.e., it is sufficiently understandable. By introducing these changes of text fragments, we reduced the repetition of phrases and similarity percentage of the entire MS. We have organized the MS in terms of the meaning and content of subsequent sections and paragraphs contained therein. Besides, current and important scientific publications in this field were used to assess and discuss the current state of knowledge in the discussed issue. They are used in the individual sections and paragraphs of the MS discussed. Please kindly check the details of these aspects in the MS.
Reviewer’s comments 3:
- The introduction is very long once compared with the other sections of the manuscript. it contains several sentences that can be deleted.
our response:
Thank you very much for this comment and your opinion.
That's right – in accordance with these comments, we changed the text: following the Reviewer's suggestions, we have revised the Introduction, removing redundant sections and/or reformulating them in a more concise form. We kindly ask you to review these individual changes in the Introduction (blue font instead of red). Please kindly check the details of these aspects in the Introduction of the MS.
Reviewer’s comments 3:
- The methodological approach is consistent, appropriate and technically sound and aligned with the main question addressed by the study.
our response:
Thank you very much for this opinion.
Reviewer’s comments 3:
- It is recommended to add some limitations of the study as this would have additional value to the manuscript, particularly enriching the discussion.
our response:
Thank you very much for this comment and your opinion.
Of course, you are right. This is, of course, a very valid point (thank you). That is why we have now added the text:
This study has several limitations. Firstly, the experiments were conducted in vitro using human plasma, which does not fully replicate the complexity of in vivo conditions. Secondly, we did not assess long‑term effects or potential interactions with other biological pathways that may influence oxidative stress in clinical settings. Thirdly, the study focused on lipid peroxidation as the primary marker of oxidative damage, without including additional biomarkers that could provide a broader picture of redox imbalance. Finally, while curcumin was tested at different concentrations, pharmacokinetic aspects such as bioavailability and metabolism were not addressed. These limitations should be considered when interpreting the findings and highlight the need for further in vivo and clinical studies. (L 376-383). Please kindly check the MS marked.
Reviewer’s comments 3:
- The authors should highlight the novelty of the study and mention what does it provide as compared with the previously published reports.
our response:
Thank you very much for this comment and your opinion.
Of course, you are right. This is, of course, a very valid point (thank you). That is why we have now added the text:
The novelty of this study lies not onLy in demonstrating the protective effects of curcumin against Ziprasidone‑induced lipid peroxidation in human plasma, but also in interpreting these findings from the perspective of cortico‑subcortical loop functioning. This approach provides a broader neurobiological context, linking oxidative stress with dopaminergic neurotransmission and the integration of cortical basal ganglia circuits, thereby offering new insights into the mechanisms underlying neuropsychiatric disorders. (L 384-388). Please kindly check the text of the end of the Introduction in the MS marked.
Reviewer’s comments 3:
- The sentences “Impairment of the antioxidant … excess of ROS.”, “These peroxides can be … (GPx) in the cytoplasm.” and “The main damage to the cells … lipids and DNA” can be supported by the following relevant and recent reference doi: 10.1080/15376516.2023.2301670.
our response:
Thank you very much for this comment and your opinion.
Of course, you are right. This is, of course, a valid point (thank you very much). That is why we have now changed the text:
original: “Impairment of the antioxidant defense system leads to an excess of ROS.”
corrected: “Impairment of the antioxidant defense system leads to an excess of ROS (Rahmouni et al., 2024).” (L 187-188).
original: “These peroxides can be neutralized by enzymatic antioxidants such as catalase (CAT), superoxide dismutase (SOD), and glutathione peroxidase (GPx) in the cytoplasm.”
corrected: “These peroxides can be neutralized by enzymatic antioxidants such as catalase (CAT), superoxide dismutase (SOD), and glutathione peroxidase (GPx) in the cytoplasm (Rahmouni et al., 2024).” (L 195-196).
original: “The main damage to the cells caused by ROS involves proteins, lipids, and DNA.”
corrected: “The main damage to the cells caused by ROS involves proteins, lipids, and DNA (Rahmouni et al., 2024).” (L 209-210).
We have added to References:
Rahmouni, F., Hamdaoui, L., Saoudi, M., Badraoui, R., & Rebai, T. (2024). Antioxidant and antiproliferative effects of Teucrium polium extract: Computational and in vivo study in rats. Toxicology Mechanisms and Methods, 34(5), 495–506. https://doi.org/10.1080/15376516.2023.2301670
Please kindly check the above sentences of the Introduction in the MS marked.
Reviewer’s comments 3:
- The conclusions are consistent with the main findings of the study, but it would be better to paraphrase it to be as a paragraph rather than highlight points.
our response:
Thank you very much for this comment and opinion.
Of course, you are right. We fully agree. This is, of course, a very valid point (thank you very much). That is why we have now changed the Conclusions:
Antioxidant properties of curcumin: in vitro studies have shown that curcumin reduces lipid peroxidation in human plasma induced by Ziprasidone. Protection against oxidative stress: the antioxidant activity of curcumin may play a role in limiting oxidative damage, which is relevant to the pathophysiology of schizophrenia and other neuropsychiatric disorders. Potential adjunctive therapeutic effect: curcumin may serve as a potential supplement in the treatment of patients receiving antipsychotic medications, heoping to mitigate adverse effects associated with oxidative stress. Need for further research: additional in vivo and clinical studies are required to confirm the efficacy and safety of curcumin as an adjunct in antipsychotic therapy. Translational relevance: findings suggest that curcumin may be applicable not only in experimental models but also in clinical contexts, such as the treatment of schizophrenia, where oxidative stress and lipid damage are key components of pathogenesis.
Our in vitro findings demonstrate that curcumin possesses antioxidant properties, effectively reducing Ziprasidone‑induced lipid peroxidation in human plasma. By limiting oxidative damage, curcumin may contribute to the protection against mechanisms relevant to the pathophysiology of schizophrenia and other neuropsychiatric disorders. These results also suggest a potential adjunctive therapeutic role for curcumin in patients receiving antipsychotic medications, where it could help mitigate adverse effects associated with oxidative stress. Nevertheless, further in vivo and clinical studies are required to confirm its efficacy and safety in this context. Importantly, the translational relevance of our findings indicates that curcumin may be applicable not only in experimental models but also in clinical practice, particularly in conditions such as schizophrenia, where oxidative stress and lipid damage are central to disease progression. (L 835-853). Please kindly check the Conclusions section in the MS marked.
Reviewer’s comments 3:
- English language is acceptable but minor checking is required.
our response:
Thank you very much for this important attention.
Of course, our MS required special linguistic attention again. We apologize for this lack of necessary corrections and thank you very much for your suggestions.
The text has now been corrected again linguistically and grammatically by the Native Speaker, and we have also re-organized it in terms of the meaning and content of subsequent sections and paragraphs.
Generally, in our corrections, we have adapted our changes to the comments of the Reviewer.We tried to use the correct logic of our arguments.
We have paid our attention to ensure that the information provided is precise and presented in a rational manner, i.e., consistent with the meaning of the aspects discussed subsequently. We have arranged individual sections according to the logical order of their content in relation to each subsequent section. Thus, individual paragraphs are related to the content of their information, in accordance with the sequence of their co-text. Please kindly check the details of these aspects in the MS.
Reviewer’s comments 3:
- Use capital letter for the unit of liter (L) throughout the whole manuscript inclusing its illustrations such as 50 µg/ml and 139 ng/ml in the main text and 5 µg/ml and 12.5 µg/ml in the figure 3 and table 1.
our response:
Thank you very much for this comment and opinion.
Of course, you are right. We fully agree. This is, of course, a valid point (thank you very much). Thus, following the Reviewer's comment, we have changed the lowercase letter for the volume unit (l) to an uppercase letter (L) throughout the manuscript, including Figures and Tables. Please kindly check these details in the MS.
Dear Reviewer 3,
Once again we thank you very much for any constructive and helpful kind opinions and suggestions, which helped us to improve our paper.
We are deeply grateful to you for your work and time spent reading and commenting our article.
We thank you very much for your understanding and patience in accordance with our any delay and irregularities.
On behalf of co-authors,
Kind regards,
Piotr Kaminski

Round 2
Reviewer 1 Report
Comments and Suggestions for Authors
The authors have addressed the reviewers’ previous comments with appropriate revisions, reflecting attention to detail and improvement of the manuscript. In its current form, the article meets the standards of quality and scientific rigor required for publication.
Reviewer 2 Report
Comments and Suggestions for Authors
After revision, there appear to be no serious issues with the manuscript, and it is recommended for acceptance.